# Snow Water Equivalent Retrieval Over Idaho, Part A: Using Sentinel-1 Repeat-Pass Interferometry

Shadi Oveisgharan[1], Robert Zinke[1], Zachary Hoppinen[2], and Hans Peter Marshall[2]

[1]Jet Propulsion Laboratory, California Institute of Technology, 4800 Oak Grove Dr, Pasadena, CA, USA
[2]Boise State University, Department of Geosciences, 1295 University Drive, Boise, ID, USA

**Correspondence:** Shadi Oveisgharan (Shadi.Oveisgharan@jpl.nasa.gov)

**Abstract.** Snow Water Equivalent (SWE) is identified as the key element of the snowpack that impacts rivers' streamflow and water cycle. Both active and passive microwave remote sensing methods have been used to retrieve SWE, but there does not currently exist a SWE product that provides useful estimates in mountainous terrain. Active sensors provide higher-resolution observations, but the suitable radar frequencies and temporal repeat intervals have not been available until recently. Interferometric Synthetic Aperture Radar (InSAR) has been shown to have the potential to estimate SWE change. In this study, we apply this technique to a long time series of 6-day temporal repeat Sentinel-1 C-band data from the 2020-2021 winter. The retrievals show statistically significant correlations both temporally and spatially with independent in situ measurements of SWE. The SWE change measurements vary between -5.3cm to 9.4cm over the entire time series and all the in situ stations. The Pearson correlation and RMSE between retrieved SWE change observations and in situ stations measurements are 0.8, and 0.93cm, respectively. The total retrieved SWE in the entire 2020-2021 time series shows SWE error less than 2cm for the 9 in situ stations in the scene. Additionally, the retrieved SWE using Sentinel-1 data is well correlated with LIDAR snow depth data with correlation of more than 0.47. Low temporal coherence is identified as the main reason for degrading the performance of SWE retrieval using InSAR data. We also show that the performance of the phase unwrapping algorithm degrades in regions with low temporal coherence. Higher frequency such as L-band improves the temporal coherence and SWE ambiguity. SWE retrieval using C-band Sentinel-1 data is shown to be successful, but faster revisit is required to avoid low temporal coherence. Global SWE retrieval using radar interferometry will have a great opportunity with the upcoming L-band 12-day repeat pass NISAR data and the future 6-day repeat pass ROSE-L data.

## 1 Introduction

The seasonal snowpack provides water resources to billions of people worldwide (Barnett et al., 2005). Snow is the primary source of water for river channel discharge in middle-to-high latitude areas. Therefore, snow mass and snow cover has a great impact on global and regional water cycles. Large scale mapping of snow water equivalent (SWE) with high resolution is critical for many scientific and economics fields. SWE is defined as the depth of water which would be obtained if all ice contained in the snow pack were melted. NASA SnowEx is a multi-year effort to improve SWE and snow surface energy

balance measurements and estimates. SWE has been identified as the key variable for terrestrial snow by the SnowEx campaign and NASA's decadal survey.

Estimating SWE on a global scale with enough accuracy and resolution is still a challenge. Passive spaceborne sensors based on the microwave emission of the snow pack (Takala et al., 2011; Kelly et al., 2003; Pulliainen and Hallikainen, 2001; Kelly, 2009) have a coarse spatial resolution on the 10 kilometer-scale. The technique saturates for SWE deeper than 150mm, which makes their application in the mountains challenging. Nevertheless, passive microwave sensors represent the current state-of-the-art of SWE retrieval methods. These sensors are applied operationally to generate daily estimates of SWE globally (Takala et al., 2011; Kelly et al., 2003), however many products such as GlobSnow mask out mountainous areas, due to the saturation limit and resolution.

Airborne LIDAR has been successful in estimating snow depth (Painter and et al., 2016). However, clouds and limited regional coverage are limiting factors for this method. This technique also needs a snow density model to estimate SWE from the LIDAR snow depth , and there currently is not a path to space for global snow depth mapping at the temporal resolution required.

Active microwave sensors provide high resolution and global coverage. There has been many efforts in the last two decades trying to estimate SWE or snow depth using active sensors mounted on a tower (Cui et al., 2016; Lemmetyinen et al., 2018; Ruiz et al., 2022; Leinss et al., 2015), airborne (Marshall et al., 2021; Nagler et al., 2022), or spaceborne systems (Lievens et al., 2019; Liu et al., 2017; Conde et al., 2019; Dagurova et al., 2020; Eppler et al., 2022). Backscattered power from active sensors is used to estimate SWE (Rott et al., 2010; Ulaby and Stiles, 1980; Cui et al., 2016; Nghiem and Tsai, 2001; Lievens et al., 2019). A dual-band (X and Ku) SAR mission has been the focus of the European Space Agency (ESA) and Canadian Space Agency (CSA) for SWE spaceborne measurements (Rott et al., 2010; Lemmetyinen et al., 2018). However, accurate *a priori* characterization of snow micro-structural parameters is of primary importance in the accuracy of SWE retrieval algorithms using backscattered power (Lemmetyinen et al., 2018; Durand and Liu, 2012; Cui et al., 2016). The most common *a priori* characterization used for SWE retrieval algorithms using backscattered power is grain radius. This has been done using passive data; however, the methods are limited by passive retrieval errors and also mismatch between active and passive resolutions. The ratio of cross-pol to co-pol Sentinel-1 backscattered power has been used to estimate snow depth over mountainous regions with deep snow (Lievens et al., 2019, 2022). Using Sentinel-1 backscattered power ratio is a unique approach showing the success of snow depth retrieval using the spaceborne radar time series data. However, the retrieval mostly works for deep snow in mountainous regions. The radiative transfer physics at C-band for this method are still poorly understood. The co-polar phase difference (CPD) between VV and HH polarization of X-band SAR acquisitions is used for estimating the depth of *fresh* snow (Leinss et al., 2014).

Lightweight and portable Frequency Modulated Continuous Wave (FMCW) radar systems have been used to map snow pack properties (such as depth, SWE, and stratigraphy) rapidly over large distances and at high resolution (Marshall and Koh, 2008). The system was normally deployed nadir looking and was a real aperture radar system. The resolution of FMCW system for SWE application is in cm scale. In order to achieve such high resolution, the bandwidth should be in GHz scale. Due to

limitation on frequency bandwidth allocation of a spaceborne active sensor (Tao et al., 2019), FMCW systems cannot be used in spaceborne missions for global coverage due to their wide bandwidth.

The phase change of specularly reflected signals in Signals of Opportunity (SoOp) is shown to be strongly dependent on SWE changes for dry snow (Yueh et al., 2017, 2021; Shah et al., 2017). The theory behind using SoOp for SWE retrieval is similar to repeat pass interferometry that is explained in section 2. The advantage of this method is that the stratigraphy of the snow has little impact on the SWE retrieval (Leinss et al., 2015; Yueh et al., 2017) similar to SWE retrieval explained in section 2. Using the long wavelength signal at P-band in SoOp is very helpful for addressing the loss of temporal coherence and

phase unwrapping challenges of this method. However, the phase sensitivity to SWE changes decreases at lower frequencies. There has been very limited data showing the success of this method at P-band. Achieving high resolution for SoOp data is another challenge (Yueh et al., 2017, 2021; Shah et al., 2017).

    As explained in details in next section, the phase difference between two SAR observations is proportional to changes in SWE variation ($\Delta SWE$). We evaluated the performance of SWE retrieval using interferometry over Idaho. In part (A) of this

study (the current paper), we used Sentinel-1 interferometric time series data over Idaho. In part (b), we will use Uninhabited Aerial Vehicle Synthetic Aperture Radar (UAVSAR) interferometric time series data over Idaho to evaluate the performance of this method. We explain SWE estimation using repeat pass interferometry in section 2. The details about different data sets used in this study are discussed in section 3. Section 4 describes how we processed Sentinel-1 data and convert it to SWE. The retrieved SWE is compared with in situ and LIDAR data in section 5. This work shows the success of SWE retrieval using a

*long* time series *spaceborne* InSAR data in winter 2021.

## 2   Using Differential Interferometry to Estimate SWE

Differential SAR interferometry measurements have been used to detect small surface elevation changes over large areas with a vertical accuracy of a few millimeters (Gabriel et al., 1989; Zebker et al., 1994). The measured phase difference is proportional and sensitive to changes in SWE variation ($\Delta SWE$) during the snow season (Guneriussen et al., 2001; H. Rott and Scheiber,

2003; Deeb et al., 2011; Leinss et al., 2015; Conde et al., 2019; Liu et al., 2017; Hui et al., 2016; Nagler et al., 2022; Eppler et al., 2022; Dagurova et al., 2020; Marshall et al., 2021). The main advantage of this method is its simplicity and a reduction in necessary *a priori* information.

    The snow volume scattering affects the interferometric phase for very deep snow in Greenland at relatively high frequencies such as C-band (Oveisgharan and Zebker, 2007). However, for the terrestrial snow, the effect of volume scattering of dry snow

on the interferometric phase is very small compared to scattering from the ground at high frequencies. The snow refractive index delays the echo received from the ground. The signal delay caused by refraction can be measured with differential radar interferometry as (Guneriussen et al., 2001; Leinss et al., 2015):

$$\Delta\phi = -2\kappa_i(\cos\theta - \sqrt{\epsilon - \sin^2\theta})\Delta d \tag{1}$$

where $\Delta\phi, \kappa_i, \Delta d, \theta,$ and $\epsilon$ are interferometric phase between two observation dates, incidence wavenumber, snow depth

change, incidence angle, and permittivity of the snow, respectively. The change in interferometric phase is used to calculate

$\Delta SWE$ (Leinss et al., 2015; Conde et al., 2019; Liu et al., 2017; Nagler et al., 2022). Similar to the dual-pol. dual-freq. retrieval algorithm (Lemmetyinen et al., 2018; Cui et al., 2016), this technique relies on the dryness of snow in order to penetrate all the way to the ground so that the scattering from the snow layers and snow volume is minimized compared to snow-ground return (Oveisgharan et al., 2020).

Using Envisat interferometric data to estimate SWE was not very successful mainly due to large temporal baseline and, hence, low temporal coherence (Hui et al., 2016). A modified version of SWE estimation using InSAR is also introduced (Eppler et al., 2022; Dagurova et al., 2020). The backscattering from the roughness in the ground and snow layers are combined with interferometric phase to improve the accuracy (Dagurova et al., 2020). The sensitivity of the dry-snow refraction-induced InSAR phase to topographic variations is used to bypass the unwrapping problem (Eppler et al., 2022). Airborne data collected

over the Austrian Alps in 2021 showed good agreement between retrieved SWE using InSAR and mean in situ SWE. Root-mean-square difference of 4.0 mm for a small snow storm of 14mm snow depth at C-band and 11.2mm for a big snow storm of 66mm at L-band were observed (Nagler et al., 2022). The correlation of 0.76 was observed between the retrieved SWE change using L-band UAVSAR differential interferometry between 2/1/2020 and 2/13/2020 and the collected LIDAR snow depth change between 2/1/2020 and 2/12/2020 over the open regions of Grand Mesa in dry snow conditions (Marshall et al.,

2021). SWE retrieval using Sentinel-1 interferometric data showed mean accuracy of 6mm over Finland for just two passes (Conde et al., 2019).

All these studies have proven the potential of this method but were limited in time or space for data collection or validation. In this study, we show the performance of SWE retrieval using a long time series of Sentinel-1 interferometric data in winter 2021. This study shows that SWE estimation using repeat pass interferometry works well by validating the retrieved value with

a large number of in situ stations and two regional LIDAR snow depth maps.

    With the recent SnowEx 2020 campaign using UAVSAR L-band differential interferometry data, Sentinel-1 C-band differential interferometry, and future NASA-ISRO SAR (NISAR) L-band data, there will be more advances in the limitations and capabilities of this method.

## 2.1   Temporal Coherence

The received radar signals at two different times will be correlated with each other if the set of scatterers in the resolution cell remain the same. However, the movement of the scatterers such as leaves and branches or sea ice particles decrease the temporal coherence (Zebker and Villasenor, 1992; Kellndorfer et al., 2022; Lavalle et al., 2012). The loss of coherence between the observations is one of the main limitations for SWE retrieval using differential interferometry. Methods such as using two frequencies or shorter revisit time are used to overcome these problems (Deeb et al., 2011; Leinss et al., 2015). Melting

and wind are the main reasons for low temporal coherence in snow (Leinss et al., 2015; Luzi et al., 2009). A medium mean temporal coherence of 0.41 is observed at L-band between two winter seasons in shrub-lands with 10.2cm average snow depth(Molan et al., 2018). Temporal coherence decreases with increasing frequency (Leinss et al., 2015; Nagler et al., 2022; Kellndorfer et al., 2022; Ruiz et al., 2022). A median temporal coherence of about 0.5 is observed at 10.2 GHz and 16.8 GHz even after 60 days (Leinss et al., 2015). However, the spaceborne TerraSAR-X temporal coherence over snow at 9.65 GHz

reduces significantly in 11 days (Leinss et al., 2015). This is probably due to random phase drifts over time that cannot be estimated and corrected in a spaceborne system compared to a ground radar. Vegetation cover decreases the temporal coherence significantly at high frequencies (Baduge et al., 2016; Kellndorfer et al., 2022; Ruiz et al., 2022). A tower-based fully polarimetric InSAR studied the effect of air temperature, precipitation intensity, and wind on the temporal decorrelation at L-, S-, C-, and X-bands (Ruiz et al., 2022). The temperature was shown to be the most critical variable affecting the temporal coherence

among other variables. Temperature above $0°C$ reduced the temporal coherence drastically (Ruiz et al., 2022). On the other hand snow cover has a thermal insulation effect on the ground and underlaying layers (Gu et al., 2019). The insulation increases with the snow depth. Therefore, during the snow season we assume the ground remains frozen even when snow becomes wet. Hence, temporal decoherence from the ground is negligible. Snow also SWE accumulation profile retrieval was successful for short temporal baselines and low frequencies in non-vegetated areas. However, the error increased for high frequencies and

long temporal baselines. The SWE profile retrieval using C-band data performs well using 12 hours and 1 day repeat-pass data. The retrieval is poor using the 12-day repeat-pass data at C-band (Ruiz et al., 2022). 6-day repeat pass C-band data showed good performance for small SWE changes but poor performance for large SWE changes between the interferometric pairs due to phase ambiguity caused by large SWE change (Ruiz et al., 2022). The low temporal coherence and low penetration depth at frequencies higher than 10 GHz, make L- and C-band desirable frequencies for differential interferometry.

**2.2    Relationship between $\Delta SWE$ and $\Delta\phi$**

With some approximation to equation 1, Leinss et al. showed a linear relationship between the interferometric phase and SWE change (Leinss et al., 2015). The approximation is limited to a smaller range of incidence angle than Sentinel-1 incidence angle. However, Leinss et al. approximation applies to a wide range of snow density up to solid ice density. Due to the wide range of Sentinel-1 incidence angle in a frame, we tried to make a more accurate approximation for a wider range of incidence angles

and snow densities limited to terrestrial snow. The snow permittivity in equation 1 is dependent on snow density, $\rho(g/cm^3)$ , and relatively independent of signal wavelength. Following Leinss et al. (Leinss et al., 2015), we use Matzler's model (Mätzler, 1987) for calculating $\epsilon$ in equation 1 ($\epsilon(\rho) = 1 + 1.5995\rho + 1.861\rho^3$ for $\rho < 0.4\frac{g}{cm^3}$; and $\epsilon(\rho) = ((1 - \frac{\rho}{0.917}) + 1.4759\frac{\rho}{0.917})^3$ for $\rho >= 0.4\frac{g}{cm^3}$ ). We can rewrite equation 1 as

$$\Delta\phi = -2\kappa_i C(\theta, \rho)\frac{\rho_{water}}{\rho}\Delta SWE \tag{2}$$

where $C(\theta, \rho) = \cos\theta - \sqrt{\epsilon(\rho) - \sin^2\theta}$. Note that the $\epsilon(\rho)$ and consequently $C(\theta, \rho)$ are unitless, $\rho_{water} = 1\frac{g}{cm^3}$, $\Delta SWE$ is in (m), and $\kappa_i$ is in (1/m).

The blue and red lines in figure 1 (a) show $C(\theta, \rho)$ versus snow density for incidence angles equal to 0 and 70 degrees, respectively. As seen in this figure, there is approximately a linear relationship between C and snow density. We fit a line to C for different incidence angles as $\hat{C}(\theta, \rho) = A(\theta) \times \rho$ for $0.15 \leq \rho \leq 0.5\frac{g}{cm^3}$. The blue and red dashed lines show $\hat{C}(\theta, \rho)$

at incidence angles equal to zero and 70, respectively. As seen in figure 1(a), the fitted line with zero intercept is a good approximation. The Zero intercept approximation is essential to retrieve $\Delta SWE$ independent of snow density. The incidence

angle mostly lies between zero and 80 for Sentinel-1 data. The terrestrial snow density lies mostly between 0.15 and 0.5 $\frac{g}{cm^3}$. Therefore, we limit ourselves to incidence angle between 0 and 80 and snow density between 0.15 and 0.45 $\frac{g}{cm^3}$ in fitting a line to C. Solid blue line in figure 1 (b) shows $A(\theta)$ versus incidence angle. By fitting a polynomial to A, we can write it as

$$\hat{A} = -0.6784\theta^2 + 0.2899\theta - 0.8473 \tag{3}$$

The dashed blue line in figure 1(b) shows the fitted curve, $\hat{A}(\theta)$. We can rewrite the equation 1 as

$$\Delta\phi = -2\kappa_i(-0.6784\theta^2 + 0.2899\theta - 0.8473)\Delta S\hat{W}E \tag{4}$$

Figure 1(c) shows the $\frac{\Delta SWE - \Delta S\hat{W}E}{\Delta SWE} \times 100$ versus incidence angle for different snow densities. As seen in this figure, the error in $\Delta SWE$ calculation using the approximation in equation 4 is less than 10% for incidence angles less than 70. We use equation 4 for estimating $\Delta SWE$ using interferometric phase, $\Delta\phi$, for the rest of this study. Using one equation for the entire Sentinel-1 frame makes the interferometric phase conversion to $\Delta SWE$ very convenient. However, we need to keep in mind that the approximation for lower dense snow has more than 10% error for incidence angle larger than 70.

## 3 Datasets

### 3.1 Sentinel-1

The Sentinel-1 radar operates at C-band at a central frequency of 5.405 GHz. It has four exclusive imaging modes with different resolutions (down to 5 m) and swath width up to 400 km. Sentinel-1 has dual polarization capability and rapid product delivery. Sentinel-1 constellation includes Sentinel-1A and Sentinel-1B. These two satellites are in the same orbit

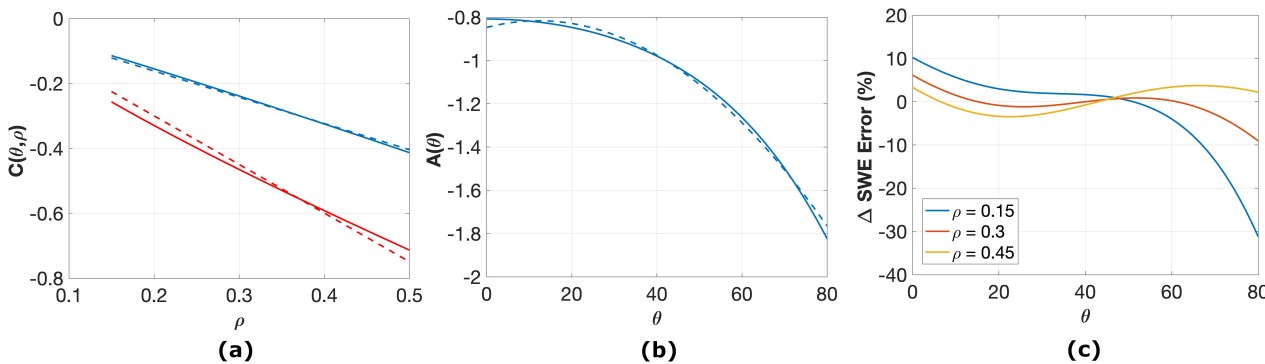

**Figure 1.** (a) $C(\theta, \rho)$ versus snow density for $\theta = 0$ and $\theta = 70$ shown by solid blue and red lines, respectively. The blue and red dashed lines show the linear fit to $C(\theta, \rho)$ with zero intercept $\hat{C}(\theta, \rho)$. (b) The line slope in part (a) versus incidence angle. The dashed line shows the fitted polynomial, $\hat{A}$ (c) $\Delta SWE$ error percentage ($\frac{\Delta SWE - \Delta S\hat{W}E}{\Delta SWE} \times 100$) assuming $\hat{C} = \hat{A}\rho$ versus incidence angle for snow density equal to 0.15, 0.3, and 0.45 $g/cm^3$, shown by blue, red, and yellow lines, respectively.

with a 180° orbital phasing difference. The revisit time for each of the satellites is 12 days. However, revisit time can get to 6 days if both satellites make observations. The data are free and available through Alaska SAR Facility (ASF) or The Copernicus Data Hub distribution service. We used the Interferometric Wide Swath (IW) mode data with 5 and 20m single look resolution in range and azimuth direction, respectively. The IW swath width is about 250km. We used ASF On Demand Processing to generate interferometric phase and coherence at vv and vh (transmit-received polarization) polarization. Alaska Satellite Facility's Hybrid Pluggable Processing Pipeline (HyP3) is a service for processing Synthetic Aperture Radar (SAR) imagery. The workflow includes interferometric phase correction for ground topography and geolocation. The ASF HYP3 uses a Minimum Cost Flow (MCF) algorithm for phase unwrapping. The unwrapped phase and interferometric coherence were used in this study. The resolution of the HYP3 phase and coherence is 80mx80m. Sentinel-1 collects data every 12 days globally but has the capability to collect the data every 6 days over targeted areas, mainly over Europe and selected areas such as SnowEx sites. In order to validate our SWE retrieval using Sentinel-1 data, we use LIDAR data from SnowEx campaign and SNOTEL data as discussed in section 5. We also use the average of SNOTEL data as a reference point for SWE retrieval, as seen in section 4.

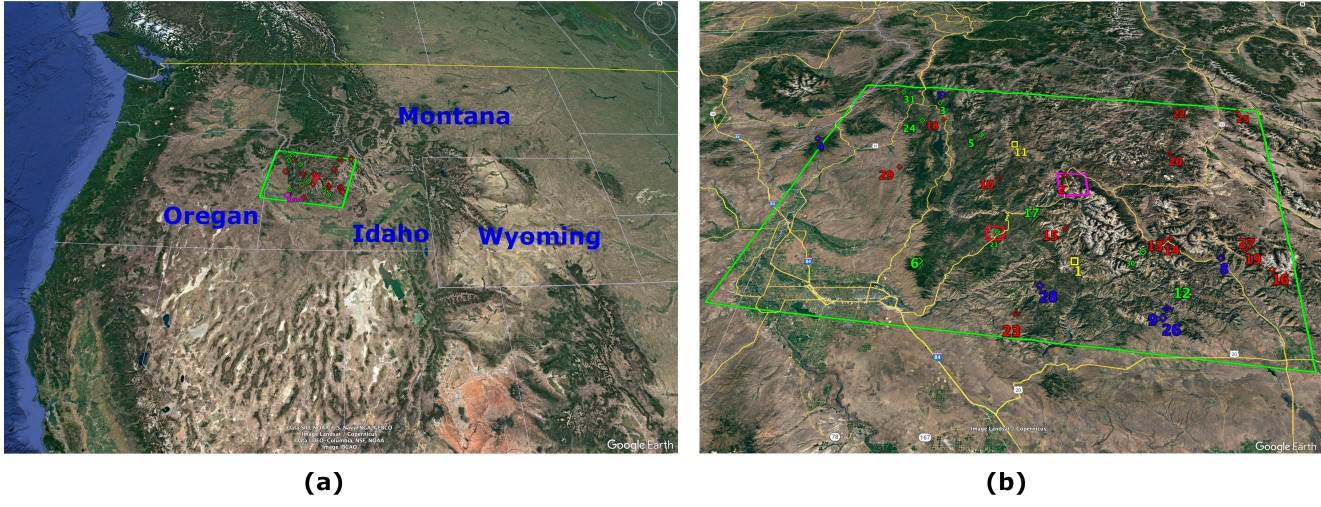

**(a)**                                                                                                          **(b)**

**Figure 2.** © Google Earth View (a) Google Earth View of Sentinel-1 path:71, frame:444 in Idaho. (b) zoomed to the Sentinel-1 path:71, frame:444, shown by big green rectangle. Red boxes show the location of LIDAR data acquisition. The green diamonds show SNOTEL stations with $\Delta SWE$ error less than 2cm in the entire time series. The red diamonds show SNOTEL stations with $\Delta SWE$ error more than 2cm in at least one observation in the time series. Yellow squares are SNOTEL stations 1 and 11 used for reference point. Blue diamonds show the location of stations with temporal coherence less than 0.35 or temperature more than $0°C$ in the entire time series.

The NASA SnowEx2021 Time Series is the continuation of the multi-year effort to improve SWE measurements and estimates. The data acquisition for different sensors and in situ collections spread over different US sites in winter 2020. These sites span a range of snow climates and conditions, elevations, aspects, and vegetation. Flight paths were designed to include

sites with ongoing snow research projects, existing ground-based remote sensing infrastructure (e.g., radar and LIDAR), snow-off and planned snow-on aerial LIDAR, and scheduled ground snow measurement. The 2021 Time Series data set covers fewer regional sites and more frequent temporal sampling compared to the 2020 campaign. The SnowEx campaign coordinated with Sentinel-1 team to observe some of the SnowEx sites with 6 days revisit during the winter, which included the Idaho SnowEx sites.

Figure 2(a) shows one of these sites that was observed every 6 days with the Sentinel-1 over Idaho. The green frame shows the geographic location of path 71, frame 444 of Sentinel-1 data. Figure 2(b) is zoomed to the Sentinel-1 frame in part (a).

## 3.2 SNOTEL

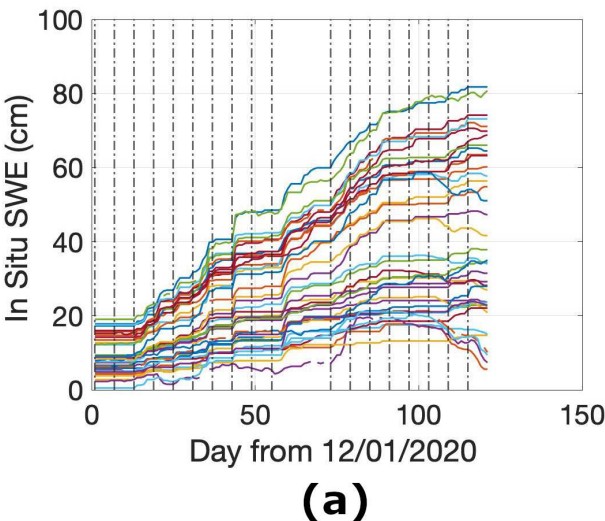
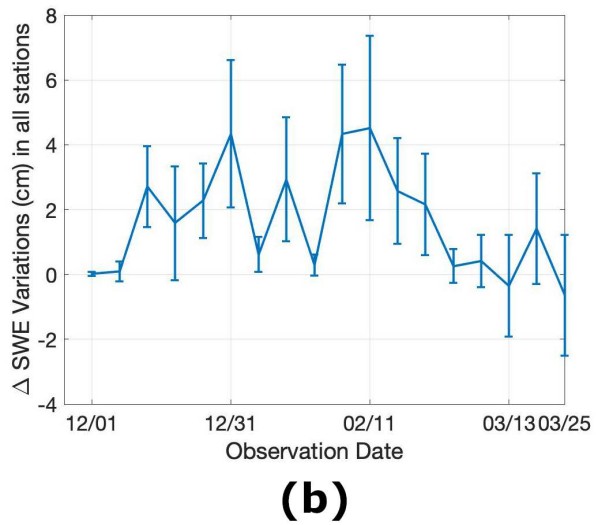

**(a)**

**(b)**

**Figure 3.** (a) The daily SWE (cm) of in situ stations shown in figure 2(b) from 12/01/20 to 03/30/21. The dashed vertical lines show the start date of Sentinel-1 observations. (b) The mean $\pm$ std of in situ $\Delta SWE$ for Sentinel-1 observation dates shown in part (a). Note that the $\Delta SWE$ is marked on the first day of each observation.

SNOwpack TELemetry Network (SNOTEL) sites are located in remote, high elevated mountainous regions in the western US. They automatically measure different snowpack characteristics and climate conditions. We used the United States Department of Agriculture (USDA) website to access hourly SNOTEL data (https://wcc.sc.egov.usda.gov/nwcc/inventory) over the region of interest shown in figure 2(b). As the Sentinel-1 frame in Idaho is collected at around 6 am local time, we downloaded the SWE, snow depth, and near surface air temperature at 6 am for each of the SNOTEL stations. Small red, green, and blue diamonds in figure 2(b) show the SNOTEL locations in Sentinel-1 frame. Figure 3 (a) shows the time series SWE of these SNOTEL sites starting from 12/1/2020 at 6am. Different colors show different SNOTEL stations. The elevation of these stations varies between 975m to 2902m. Therefore, the large spread of SWE between different stations in figure 3(a) is expected.

The dashed vertical lines are the start date of each 6-day repeat Sentinel-1 data. As seen in this figure, there is a one 6-day repeat data acquisition gap in Sentinel-1 data on 2/5/21. Figure 3 (b) shows the mean $\pm$ std of SNOTEL $\Delta SWE$ between the start date Sentinel-1 data in figure 3(a) and 6 days later. We used the SWE data from these in situ stations for (a) SWE retrievals validation by comparing retrieved $\Delta SWE$ with SNOTEL $\Delta SWE$(as seen in section 5.1), and (b) the InSAR reference point by subtracting the average of two SNOTEL $\Delta SWE$ from the retrieved $\Delta SWE$ (as explained in section 4).

### 3.3  QSI LIDAR

Airborne LIDAR provides high-resolution snow depth maps. These data are reliable sources of validation data, particularly a powerful constraint for InSAR retrieval of SWE. We used the LIDAR data for validating the retrieved SWE results. The "SnowEx20-21 QSI LIDAR DEM 0.5m" data set is part of the SnowEx 2020 and SnowEx 2021 campaigns (Adebisi et al., 2022). The data includes digital elevation models, snow depth, and vegetation height with 0.5m spatial resolution. Data were acquired over multiple areas in Colorado, Idaho, and Utah during February 2020, March 2021, and September 2021. The two red boxes in figure 2(b) show the location of LIDAR data acquisition. The big purple box is over Banner Summit and the small red box is over Mores Creek in Idaho. Figures 10(a) and 11(a) show the QSI snow depth over Banner Summit and Mores Creek, respectively. We used this data in section 5.2 to compare with retrieved SWE using Sentinel-1 data.

## 4  SWE Retrieval Using Sentinel-1 interferometric Phase

As mentioned in section 3.1, Sentinel-1 data were collected every 6 days over the region shown in figure 2(b) during 2020 and 2021, following coordination between the SnowEx campaign and the Sentinel-1 team. We used 6-day repeat Sentinel-1 time series data between 12/1/20 to 3/30/21. We selected this period to (a) capture most of the seasonal snow storm and (b) avoid wet snow as much as possible. The main sources of error in the science and applications using Sentinel-1 repeat-pass interferometry are (1) tropospheric noise, (2) temporal decorrelation, and (3) phase ambiguity. We removed tropospheric noise from the unwrapped phase as explained in section 4.1. The unwrapped phase is converted to $\Delta SWE$ using equation 4. Temporal decorrelation is relatively high at C-band. The 6-day repeat time improves the temporal coherence significantly over snow compared to the normal 12-day Sentinel-1 repeat time. In this study, any pixel with temporal coherence more than 0.35 is considered reliable. Temporal coherence of 0.35 is arbitrary, but based on experience working with InSAR data, it is a reasonable threshold number. However, for the results in section 5.2, we used all the time series data, including the data with low coherence, to calculate total SWE. The reason is that in order to compare the total SWE on a date close to LIDAR acquisition date, we need the whole $\Delta SWE$ time series up to that date.

Phase ambiguity is still one of the big sources of error in some of our data as discussed in section 5.1.2. The radar signal propagating through ionosphere is delayed. The delay is a function of frequency of the signal, Earth's magnetic field, and total electron content (TEC) and affects the accuracy of $\Delta SWE$ retrieval. The ionospheric error at C-band is much smaller than other sources of error and we consider it negligible in this study.

The temperature is also an important factor. Equation 1 is valid for dry snow (Leinss et al., 2015), and we use near surface air temperature above $0°C$ as a metric that indicates wet snow in snow season. Any retrieved SWE with SNOTEL near surface air temperature more than $0°C$ is unreliable in our study. Similar to coherence filtering, for the results in section 5.2, we used all the time series data, including the data with temperature more than $0°C$. Similar to temporal coherence, the reason is that

240    in order to compare the total SWE with LIDAR snow depth at LIDAR acquisition date, we need the entire $\Delta SWE$ time series up to that date.

Another important factor in interferometric phase images is the reference point to calibrate the unwrapped phase or consequently $\Delta SWE$. In geophysics applications using InSAR, the reference point is a stable target with no displacement or known displacement in the time interval between acquisition of the two images. For $\Delta SWE$ estimation using InSAR, the reference

245    point is chosen either by corner reflectors (cleaned of snow) with stable zero phase (Nagler et al., 2022; Dagurova et al., 2020) or using the average of in situ $\Delta SWE$ (Conde et al., 2019) or using a snow free region (Tarricone et al., 2023). As seen in figure 2(b), there are large number of in situ stations in this frame. In this study, we used the average of two in situ $\Delta SWE$ to calibrate the retrieved $\Delta SWE$ images. The two selected in situ stations have reliable measurements (coherence more than 0.35

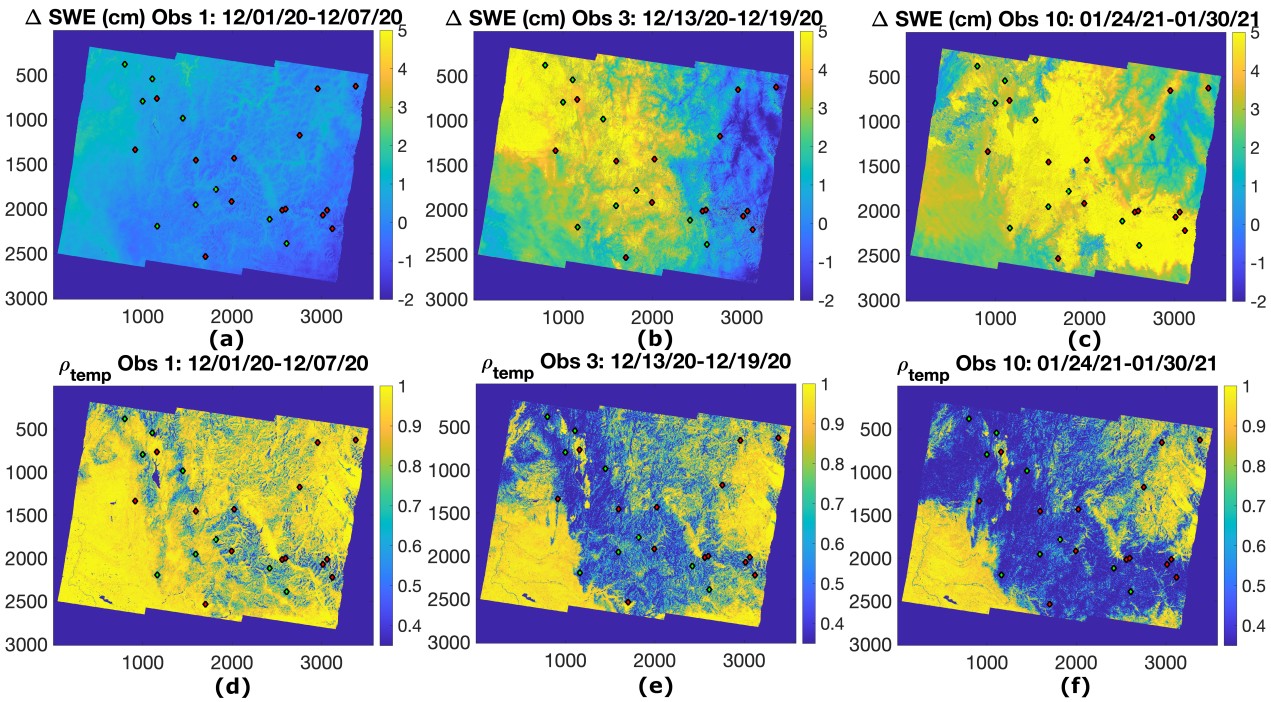

**Figure 4.** Retrieved $\Delta SWE$ using Sentinel-1 path:71, frame:444 interferometric phase data between (a) 12/01/20 and 12/07/20 (b) 12/13/20 and 12/19/20 (c) 01/24/21 and 01/30/21. Sentinel-1 path:71, frame:444 coherence between (d) 12/01/20 and 12/07/20 (Observation 1) (e) 12/13/20 and 12/19/20 (Observation 3) (f) 01/24/21 and 01/30/21 (Observation 10).The small diamonds are in situ locations. The average of in situ $\Delta SWE$, for images (a), (b), and (c) are 0.01cm, 2.72 cm, and 4.33 cm, respectively.

and temperature less than $0°C$ ) for the entire time series. For the rest of this study we used in situ stations 1 and 11 $\Delta SWE$ values to calibrate the retrieved $\Delta SWE$. Stations 1 and 11 are shown by yellow squares in figure 2(b).

Figures 4(a), (b), and (c) show retrieved $\Delta SWE$ between 12/1/20 and 12/7/20, 12/13/20 and 12/19/20, and 1/24/21 and 1/30/21, respectively. The small diamonds show the location of in situ stations in this Sentinel-1 frame. The average of in situ $\Delta SWE$, for images 4(a), (b), and (c) are 0.01cm, 2.72 cm, and 4.33 cm, respectively. The retrieved $\Delta SWE$ images in top row of figure 4 show no SWE change in part (a) and snow storms in part (b) and (c) which match the in situ measurements.

Bottom row of figure 4 shows the coherence of the images in top row of figure 4. Interferometric decorrelation has different sources, such as temporal decorrelation, volume decorrelation, signal to noise ratio decorrelation, geometric decorrelation, .... The volume decorrelation is negligible due to relatively small Sentinel-1 perpendicular baseline. Temporal decorrelation is the dominant source of decorrelation. For the rest of this study, we assume the observed interferometric decorrelation is approximately the temporal coherence. As shown in figures 4(e) and (f), snow storms reduce the coherence significantly whereas no SWE change shows very small decorrelation, as expected.

## 4.1 Tropospheric Noise Removal

A radio wave's differential phase delay variation through the troposphere is one of the largest error sources in Interferometric Synthetic Aperture Radar (InSAR) measurements, and water vapor variability in the troposphere is known to be the dominant factor. The differential delay present in a given interferogram may reach tens of centimeters. Various ways of mitigating tropospheric effects are routinely employed. Here, we used a global atmospheric weather model to predict the radar phase delay due to variations in atmospheric pressure and water vapor content between passes. Specifically, we used the European Center for Medium-Range Weather Forecasts (ECMWF) ERA5 model of atmospheric variables, which provides hourly estimates on a 30 km global grid based on assimilation of surface and satellite meteorological data. We used the Python-based Atmospheric Phase Screen (PyAPS) software (Jolivet et al., 2011) to interpolate this grid, and convert those variables into a radar phase delay. PyAPS is integrated into, and leveraged by, the Miami InSAR Time-series software in Python (MintPy) (Yunjun et al., 2019). We used MintPy to crop the atmospheric delays to match the spatial extent of the interferograms, and projected the delays into radar line-of-sight (LOS). It should be noted that while the ERA weather models often provide a reliable method for representing atmospheric phenomena at $> 30 - km$ wavelengths (grid spacing), they are less accurate at finer spatial scale, where atmospheric conditions can vary as a function of topography. Model interpolation between grid nodes as a function of elevation were performed, however some over-smoothing of atmospheric variations might still occur. More work is necessary to better determine the overall effectiveness of atmospheric phase removal, including whether tropospheric delay is completely mitigated or over-corrected, and on what spatial scales.

Figure 5 shows an example of how significant tropospheric noise can be in an InSAR image. Figure 5(a) shows the line of sight (LOS) displacement with no atmospheric correction over our area of interest in figure 2(b) between 03/13/21 and 03/19/21. Figure 5(b) shows the atmospheric noise estimation using PyAPS. Figure 5(c) shows LOS displacement after tropospheric noise removal by subtracting 5(b) from 5(a). Comparing figures 5(a) and (c), we can see that the atmospheric noise can affect the estimated $\Delta SWE$ by 5-10cm (LOS displacement error converted to $\Delta SWE$) in upper left of the images.

# 5    Results And Discussions

In this section we compare retrieved SWE using Sentinel-1 interferometric phase with in situ stations and LIDAR data.

## 5.1    Comparing Retrieved SWE using Sentinel-1 and SNOTEL SWE

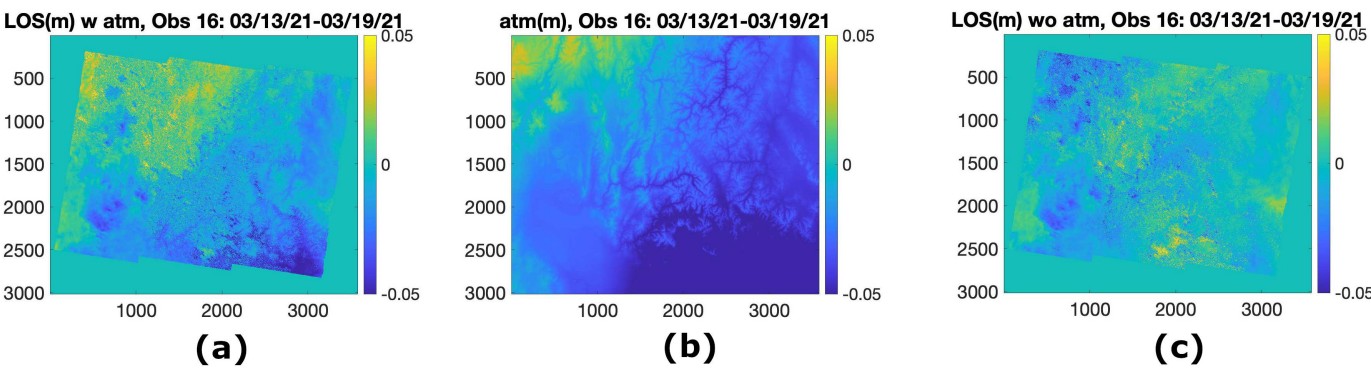

**Figure 5.** Sentinel-1 path:71, frame:444 (a) Line of Sight (LOS) displacement (m) With atmospheric noise (b) Atmospheric noise (m) (c) Line of Sight displacement (m) Without atmospheric noise, between 03/13/21 and 03/19/21.

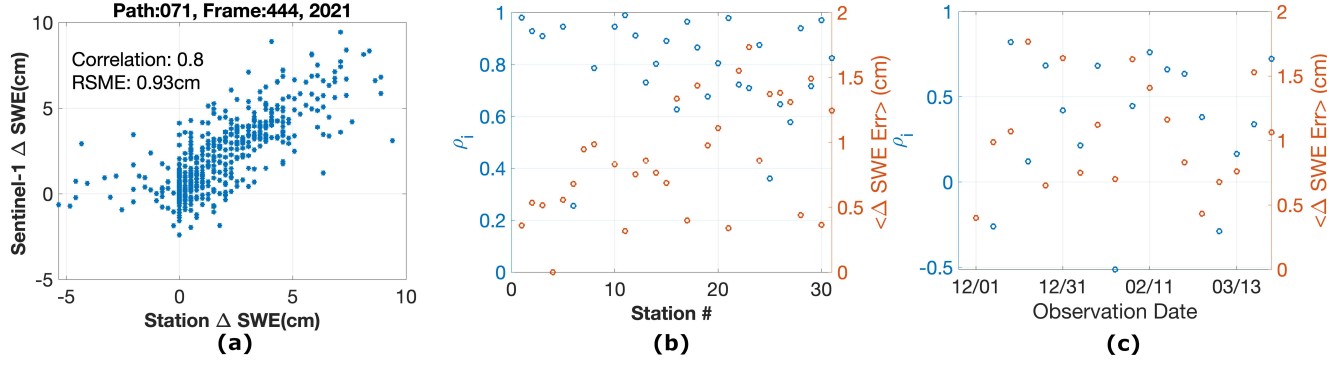

**Figure 6.** (a) Retrieved $\Delta SWE$ using Sentinel-1 interferometric phase versus in situ $\Delta SWE$ for all the stations with temporal coherence more than 0.35 for the entire Sentinel-1 time series from December 2020 to March 2021. (b) Correlation (left axis) and absolute error (right axis) between retrieved $\Delta SWE$ using Sentinel-1 interferometric phase and in situ $\Delta SWE$ for each in situ station. (c) Correlation (left axis) and absolute error (right axis) between retrieved $\Delta SWE$ using Sentinel-1 interferometric phase and in situ $\Delta SWE$ for each interferogram. Note that the labels on x-axis show the first date of each interferometric observation.

### 5.1.1 Comparing Retrieved $\Delta SWE$ using Sentinel-1 and SNOTEL $\Delta SWE$

We used all the retrieved $\Delta SWE$ (using the Sentinel-1 data from 12/1/20 to 3/30/21) for in situ stations shown in figure 2(b) and compared them with corresponding SNOTEL $\Delta SWE$. As mentioned in section 4, any retrieved value with temporal coherence less than 0.35 and temperature higher than $0°C$ is discarded. Note that the data shown in figure 6 is the SWE *change* between two consecutive Sentinel-1 data that are 6 days apart. We showed the $\Delta SWE$ for all stations and all consecutive observations between 12/1/20 and 3/30/21. As mentioned in section 3.1, the resolution of the Sentinel-1 InSAR data from Hyp3 is 80mx80m. We used a 10x10 multi-looks window of retrieved SWE and temporal coherence around the SNOTEL locations to reduce the speckle noise. Therefore, we compared the SNOTEL SWE with the 800mx800m retrieved SWE around the SNOTEL site. The heterogeneity of the environment such as vegetation cover, vegetation fraction, land type, and SWE distribution in the 800mx800m around the SNOTEL station affects our accuracy. We will analyze the effect of the heterogeneity of the environment on the SWE retrieval for SNOTEL stations in the future work of this study.

Figure 6 (a) compares all the retrieved $\Delta SWE$ time series using Sentinel-1 data over all in situ stations with SNOTEL $\Delta SWE$. As seen in this figure, the retrieved and in situ $\Delta SWE$ are highly correlated (0.8), with an RMSE of 0.93cm .

Figure 6(b) shows the correlation and RMSE between the entire time series of retrieved and in situ $\Delta SWE$ for each station, by blue and red circles respectively. As seen in this figure, the correlation is good (more than 0.6 for all stations except three). The RMSE is less than 2cm for all stations and less than 1cm for most stations. Note that station 4 has just one observation with temporal coherence more than 0.35. That observation is the first observation with zero SWE change. Therefore, there is not enough points to calculate $\rho_i$. Hence, RMSE and correlation are zero.

Figure 6(c) shows the correlation and RMSE between the in situ stations and retrieved $\Delta SWE$ for each Sentinel-1 acquisition first date, by blue and red circles respectively. Note that the labels on x-axis show the first date of each interferometric observation. The RMSE is again less than 2cm for all dates and less than 1cm for many dates. As seen in this figure, the correlation is more than 0.4 for some dates and poor (less than 0.4) for some others. Among the observation dates with correlation less than 0.35 (observation 1, 2, 4, 7, 9, 15, 16, and 17), observations 1, 2, 7, 9, 15, and 16 (first date of 12/01, 12/07, 01/06, 01/18, 03/07, and 03/13) have very small snow accumulation (the average $\Delta SWE$ is less than 0.5cm with $\Delta SWE$ close to zero for most stations). Therefore, the phase is not sensitive enough to SWE change, hence low correlation. For observation 4 and 17 (first date of 12/19, and 03/19), we observed that the low coherence degrades the phase unwrapping performance for these InSAR images. Figure 7(a) and (b) show the wrapped phase for observations 4 and 5, respectively. Note that the correlation between in situ and retrieved $\Delta SWE$ in figure 6(c) is 0.1 for observation 4 and 0.7 for observation 5. The average in situ $\Delta SWE$ between 12/19/20 and 12/25/20 (observation 4) is 1.6cm and between 12/25/20 and 12/31/20 (observation 5) is 2.3cm. However, the interferometric fringes in figure 7(a) are very noisy compared to figure 7(b). We observe that 4 out of 6 days between 12/19/20 and 12/25/20 (observation 4) are relatively warm including day 12/19/20. All 31 stations have temperature between $-7°C$ to $6°C$ at 6 am in those four days. The warm days cause a lot of melting and refreezing in those 4 days. Hence, we expect to have small temporal coherence and consequently noisier fringes. On the other hand, the temperature is relatively warm only on 12/26/20. The rest of the 5 days between 12/25/20 and 12/31/20 (observation 5) are mostly colder than

320  $-7°C$ for all 31 stations. Therefore, higher temporal coherence and less noisier fringes. We believe the noisy fringes degrade the performance of the unwrapping algorithm significantly. Therefore, the retrieved $\Delta SWE$ is more accurate for observation 5 compared to observation 4. One of the main future works of this study is to improve the phase unwrapping over images with low coherence.

### 5.1.2  Comparing Retrieved Total SWE using Sentinel-1 and SNOTEL Total SWE

325  In this section, we used time series retrieved $\Delta SWE$ to calculate total SWE at each date compared to start date of our time series (12/01/2020) by

$$SWE(t_{i+1}) = \sum_{t_j=t_1}^{t_i} \Delta SWE(t_j, t_{j+1}) \tag{5}$$

where $t_1$ is 12/01/2020. For instance, SWE at 12/25/20 compared to 12/01/2020 is the summation of all four retrieved $\Delta SWE$ ($\Delta SWE_{12/01/20-12/07/20} + \Delta SWE_{12/07/20-12/13/20} + \Delta SWE_{12/13/20-12/19/20} + \Delta SWE_{12/19/20-12/25/20}$). Note
330  that the $SWE(t_{i+1})$ is measured compared to $SWE(t_1)$. For simplicity, we assume the SWE at time $t_1$ is equal to zero.

Figures 8 (a), (b), and (c) show the time series of total SWE for in situ stations 12, 30, and 20, respectively. Note that we used average of stations 1 and 11 $\Delta SWE$ for reference point in this study. The red and blue lines show the retrieved and in situ total SWE at each Sentinel-1 date acquisition compared to 12/01/2020. However, as mentioned in section 4 and 5.1.1, we

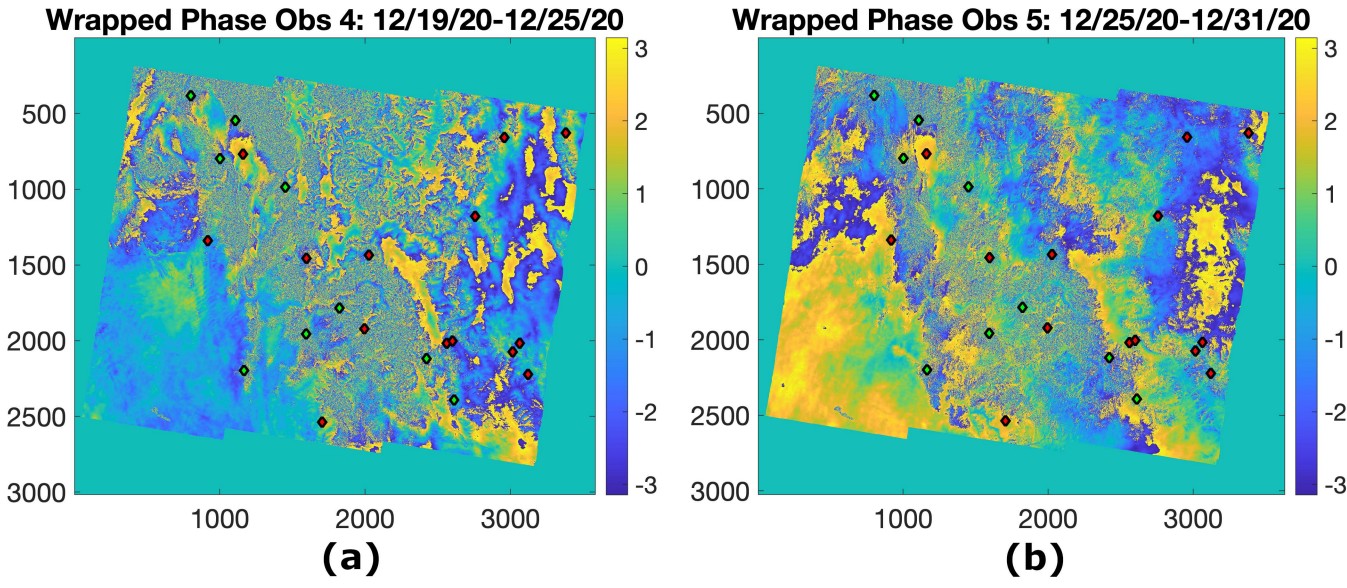

**Figure 7.** Sentinel-1 Wrapped Phase path:71, frame:444 between (a) 12/19/20 and 12/25/20 (Observation 4) (b) 12/25/20 and 12/31/20 (Observation 5).

only used $\Delta SWE$ values with temporal coherence more than 0.35 and temperature less than $0°C$. We had 18 observations
for the entire time series. Discarding some observation due to low temporal coherence or high temperature, changes the time
series length. As seen in figure 8, we keep all 18 observations for station 20 but only 15 observations for station 12.

As seen in this figure, the time series of total retrieved SWE aligns closely with in situ values for stations 12 and 30. The error
is less than 2cm in the entire time series. However, the retrieved SWE for station 20 diverges from in situ values even though it
follows the same pattern. The error in total SWE estimation is about 10cm at the end of the time series. We think the main reason
for divergence is the phase unwrapping error and phase ambiguity. As discussed in section 5.1.1, the noisy fringes degrade the
performance of the unwrapping algorithm. A similar problem is observed in tower-based studies. The retrieval diverges from
the in situ values by phase ambiguity values over large snow storms at C-band (figure 13(c) in (Ruiz et al., 2022)). However,
even in these cases, the trends of SWE remain the same between retrieved and in situ values. We will investigate the reason
behind the divergence of retrieved SWE from in situ SWE of these stations in the future work of this study.

Figure 9 (a) shows the Sentinel-1 $\Delta SWE$ ambiguity versus incidence angle. The red line shows the $\Delta SWE$ ambiguity
using equation 4 ($\Delta\phi = 2\pi$). The blue line shows the $\Delta SWE$ ambiguity using Leinss et al. approximation ($\Delta\phi = \kappa_i(1.59 +$
$\theta^{2.5})\Delta SWE$) (Leinss et al., 2015). As seen in this figure, $\Delta SWE$ ambiguity is between 1.5 to 3.5 cm depending on the
incidence angle. The relatively small $\Delta SWE$ ambiguity of Sentinel-1 makes the unwrapping challenging for snow storms.
Figure 9(b) shows the temporal coherence between 02/11/21 and 02/17/21. We can see very low coherence in the snow storm
regions which degrades the unwrapping process. As mentioned before, one of the main future projects of this study is to work
on improving the unwrapping phase.

For each station plot in figure 8, we also report the average RMSE error ($< \Delta SWEErr_{station\#} >$) and correlation ($\rho_{station\#}$)
between retrieved and in situ $\Delta SWE$, as also plotted in figure 6(b). We also report the average of temporal coherence for all
the interferograms over that station ($< \rho_{temp} >$) to show how reliable the measurements at that station are. For all three sta-
tions, the RMSE error for $\Delta SWE$ is less than 1.1 cm, the correlation between in situ and retrieved $\Delta SWE$ is greater than

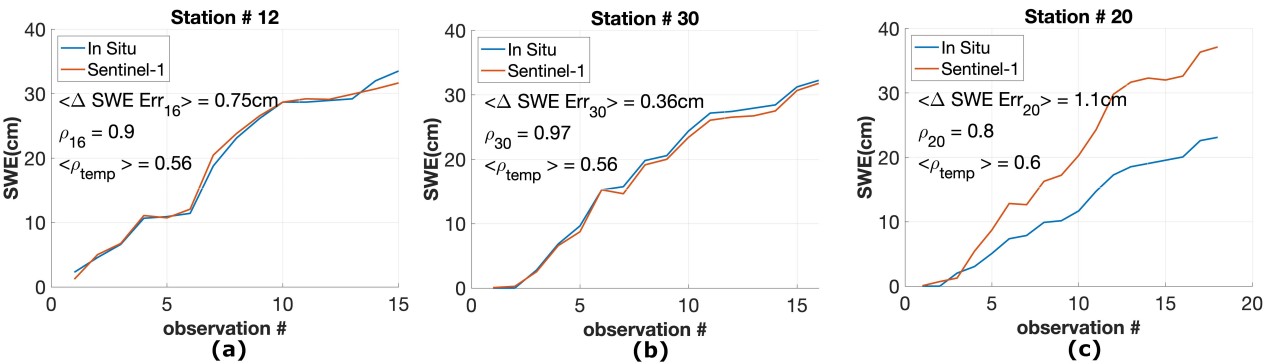

**Figure 8.** Time series of total in situ and retrieved SWE using Sentinel-1 interferometric phase shown by blue and red lines, respectively for
stations 12 (shown in 8(a)), 30 (8(b)) and 20 (8(c)).

0.8, and temporal coherence is greater than 0.5. The SNOTEL sites are shown by small diamonds in figure 2(b). The green small diamonds have a total SWE error less than 2 cm in the entire time series, similar to stations 12 and 30. The red diamonds have a total SWE error more than 2cm, similar to station 20. However, the retrieved SWE has a similar pattern as in situ SWE. Therefore, we think they have a phase unwrapping problem similar to station 20. These stations are also shown in figure 7(a). As seen in this figure, the red diamonds are mostly located in regions with noisy fringes which makes the unwrapping challenging. Among all 31 stations in the Sentinel-1 frame, 6 of them have temporal coherence less than 0.35 or temperature more than $0°C$ in their entire time series. These stations are shown by blue diamonds in figure 2(b). Two stations are used for calibration of the phase. Hence these two stations cannot be used for comparisons. So, there were 23 stations with more than 2 reliable observation dates in their time series. Among the 23 stations, 9 have SWE error less than 2cm (green diamonds) and 14 of them have SWE error larger than 2cm (red diamonds).

## 5.2 Comparing Retrieved SWE using Sentinel-1 and LIDAR SWE

As mentioned in section 3.3, the QSI LIDAR data were collected during the SnowEx campaign. There are two LIDAR data sets collected over the Sentinel-1 path:71, frame:444 in winter 2021. The locations are shown with red rectangles in figure 2(b).

Figures 10(a) and 11(a) show the LIDAR snow depth on 3/15/21 over Banner Summit and Mores Creek, respectively. As shown in figure 2(b), Banner Summit covers SNOTEL 2 and Mores Creek covers SNOTEL 21. These two SNOTEL stations

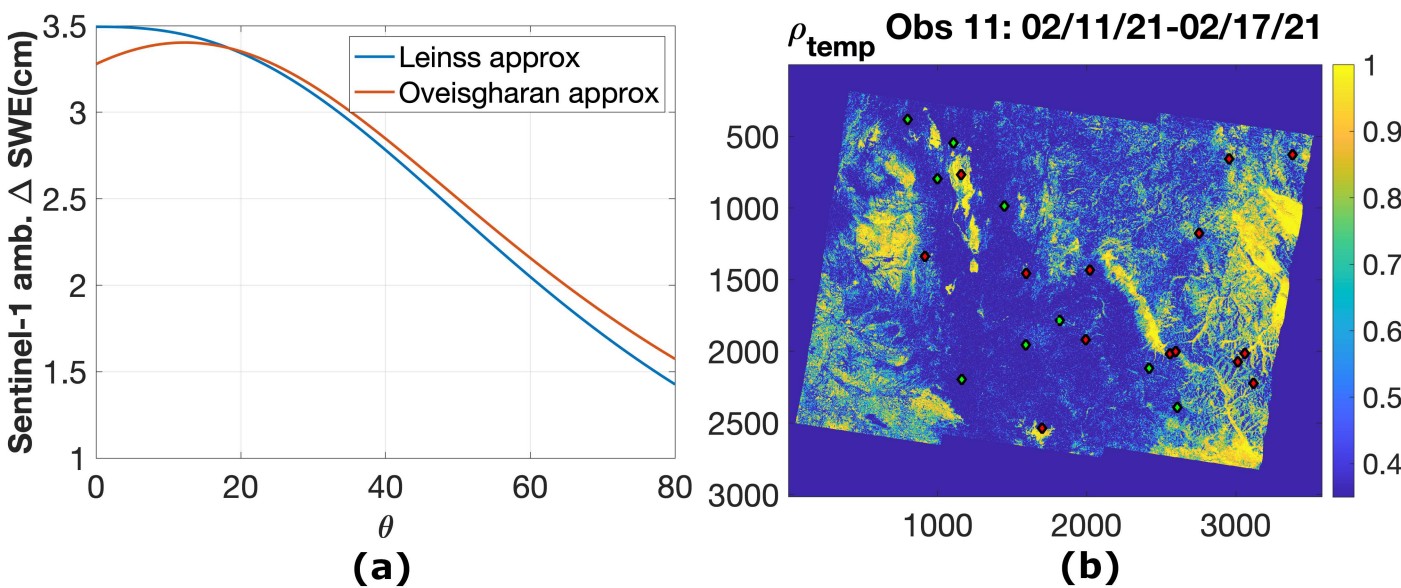

**Figure 9.** (a) $\Delta SWE$ ambiguity versus incidence angle using Leinss's approximation (blue line) and Oveisgharan's approximation (red line) (b) coherence for data acquired between 02/11/21 and /02/17/21 (Observation 11). Green diamonds show the location of stations with less than 2 cm total SWE error. Red diamonds show the location of stations with more than 2 cm total SWE error.

are shown by diamonds in figures 10(a) and 11(a). The terrain DEM is measured by LIDAR sensor during September 2021. The DEM is used to measure the snow depth using the LIDAR data collected on 3/15/2021. The big purple rectangle in figure 2(b) corresponds to Banner Summit and small red rectangle corresponds to Mores Creek. We calculated the total SWE compared to 12/01/2020 on the closest day to LIDAR date acquisition. We used all the retrieved $\Delta SWE$ from 12/01/2020 to 03/19/2021,

and calculated the total SWE on 03/19/21 using equation 5. Figures 10(b) and 11(b) show the retrieved SWE on 03/19/21 over Banner Summit and Mores Creek, respectively. The part (a) and (b) in figures 10 and 11 have very similar patterns. The 2D-histograms of these two images are shown in figure 10(c) and 11(c) where x- and y-axis show the LIDAR snow depth and Sentinel-1 retrieved SWE, respectively. The colors in part (c) shows the $10^{number\ of\ cells}$ with LIDAR snow depth x and InSAR SWE y. The correlation between these two data sets is 0.47 for Banner Summit and 0.59 for Mores Creek. Note that

the LIDAR data show the snow depth whereas Sentinel-1 retrieved data show the total SWE accumulated during the Sentinel-1 overpasses analyzed. On the other hand, LIDAR has a much higher resolution. The relatively good correlation (0.47 and 0.59) between the two independent measurements with different resolutions is a very good indication of the success of this method in estimating SWE.

Figures 12 (a) and (c) show the mean of all Sentinel-1 temporal coherence data between 12/01/20 to 03/19/21 over Ban-

ner Summit and Mores Creek, respectively. As seen in these figures, the temporal coherence varies between 0.2 to 0.9. As mentioned earlier in this section, the correlation between LIDAR snow depth data on 03/15/21 and retrieved total SWE using Sentinel-1 data on 3/19/21 is 0.47 for Banner Summit and 0.59 for Mores Creek. However, some of the points may have low temporal coherence and not viable for retrieval as discussed in section 4. Left axis in figures 12 (b) and (d) show the correlation between LIDAR snow depth data on 03/15/21 and retrieved total SWE using Sentinel-1 data on 3/19/21, for points with mean

temporal coherence above $\rho_{temp,threshold}$. Figure 12(b) shows the correlation versus $\rho_{temp,threshold}$ over Banner Summit and figure 12(d) shows the correlation over Mores Creek. Note that the correlation is 0.47 for Banner Summit and 0.59 for Mores Creek with no filter ($\rho_{temp,threshold} = 0.1$) as reported in figure 10(c) and 11(c), respectively. Right axis in figure 12(b) and

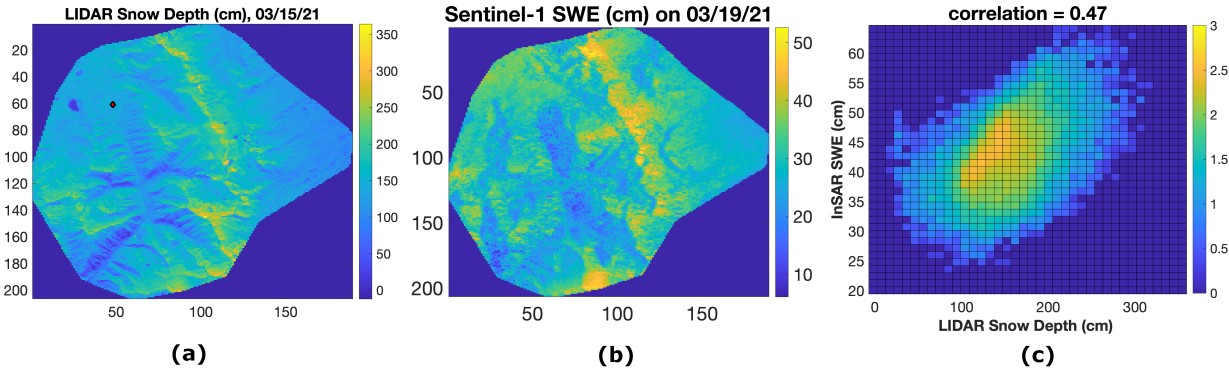

**Figure 10.** (a) QSI LIDAR snow depth over Banner Summit, ID on 3/15/21 (b). Retrieved total SWE using Sentinel-1 interferometric data from 12/1/20 to 3/19/21 over Banner Summit, ID. (c) 2D histogram of data in (b) versus data in (a).

(d) show the number of points in the image in part (a) and (c) with mean temporal coherence more than $\rho_{temp,threshold}$. There are 5 times more points in figure 12(a) compared to figure 12(c). Therefore, we can better do statistical evaluation for part (b)

compared to part (d). As seen in figure 12(b), the correlation between LIDAR snow depth and retrieved total SWE increases by filtering out points with low temporal coherence, as expected. We need to investigate more to explain the reason for correlation decrease in $0.74 < \rho_{temp,threshold} < 0.8$ interval.

The correlation between LIDAR snow depth and retrieved total SWE in figure 12(d) is relatively constant with increasing $\rho_{temp,threshold}$. However, as we increase the $\rho_{temp,threshold}$ to more than 0.46, the correlation gradually decreases to 0.4 at

$\rho_{temp,threshold} = 0.65$ and remains relatively constant up to $\rho_{temp,threshold} = 0.72$. The number of points in the image with temporal coherence more than 0.72 is less than 20. Therefore, the correlation is not statistically very meaningful. Mores Creek has a lower elevation (6100m at station 21) compared to Banner Summit (7040m at station 2). Mores Creek is also warmer (mean temperature of $-4.3°C$ for the entire time series at station 21) than Banner Summit (mean temperature of $-8.8°C$ for the entire time series at station 3). We expect to have higher correlation with filtering low temporal coherence points as seen

in figure 12(b). We think the reason we don't see such a behavior in figure 12(d) is that the warmer temperature , melting and refreezing degrade the retrieval performance even for highly correlated regions. More investigation is needed to better explain the constant or decreasing correlation with increasing $\rho_{temp,threshold}$ in figure 12(d).

Left axis in figure 13(a) and (b) show the correlation between LIDAR snow depth on 03/15/21 and retrieved total SWE for each observation between 12/01/20 and 03/19/21 over Banner Summit and Mores Creek, respectively. Observation 16 shows

the correlation reported in figure 10(c) and 11(c). Right axis in figure 13(a) and (b) show $\Delta SWE$ (cm) for each observation between 12/01/20 and 03/19/21 at station 2 in Banner Summit and station 21 in Mores Creek, respectively. As seen in both figures, the correlation gradually increases after observation 7 or 8, as expected. On the other hand, the correlation is smaller for observation 16 compared to 15. Observation 16 shows the total SWE on 03/19/21 and LIDAR data shows the snow depth on 03/15/21. There is about 1 cm $\Delta SWE$ for observation 16 that is not fully captured by LIDAR. Therefore, comparing

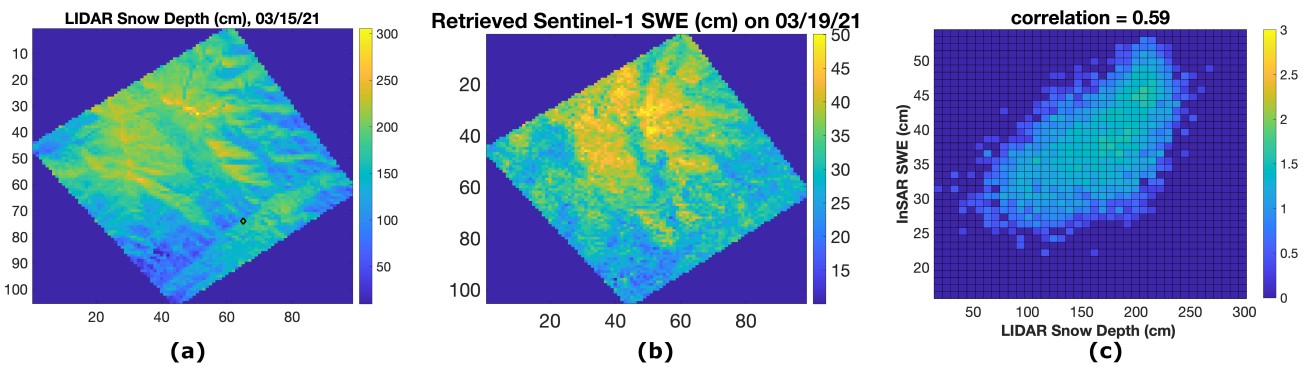

(a)                  (b)                  (c)

**Figure 11.** (a) QSI LIDAR snow depth over Mores Creek, ID on 3/15/21. (b) Retrieved total SWE using Sentinel-1 interferometric data from 12/1/20 to 3/19/21 over Mores Creek, ID. (c) 2D histogram of data in (b) versus data in (a).

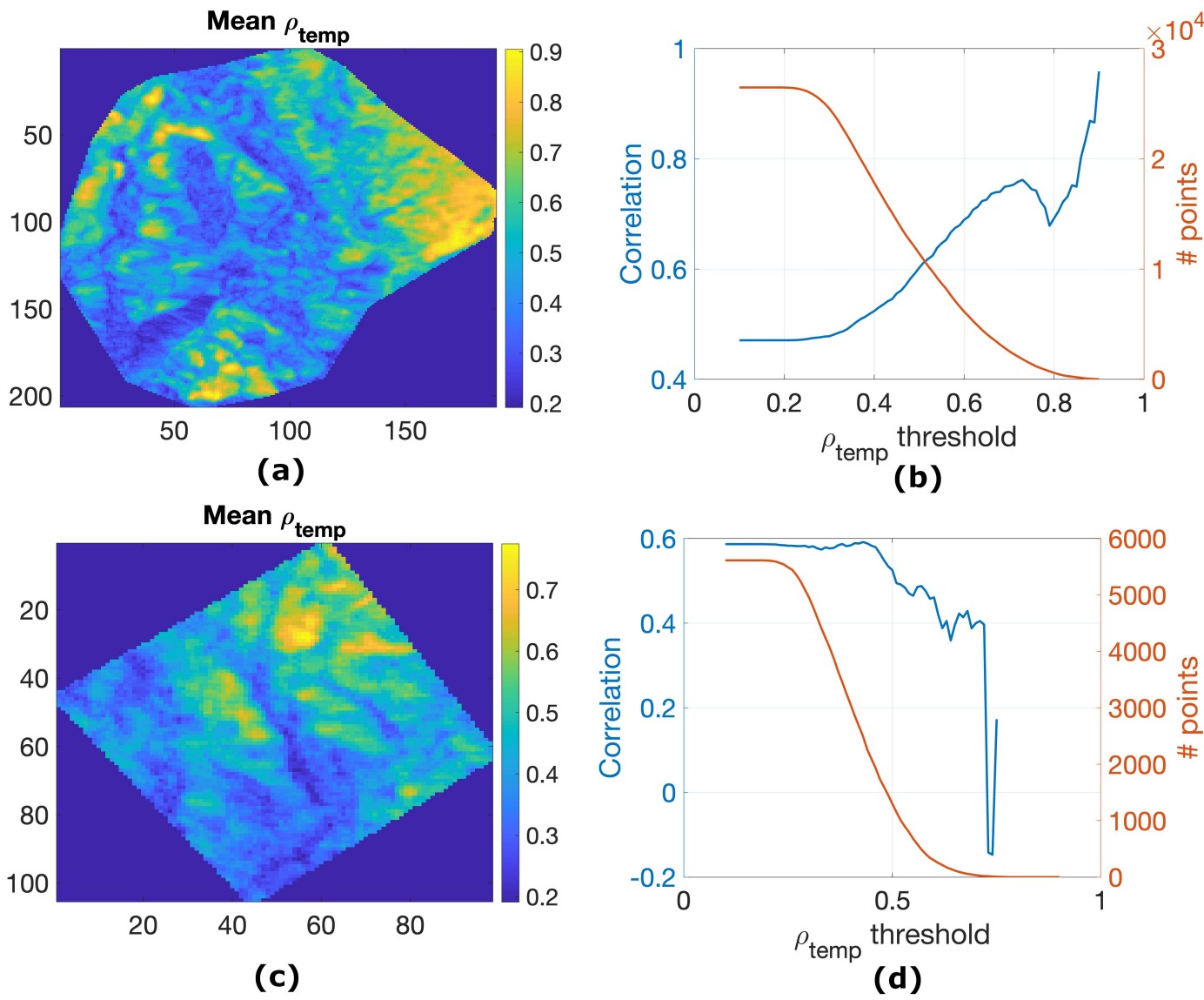

**Figure 12.** (a) Mean of Sentinel-1 temporal coherence between 12/01/20 to 03/19/21 over Banner Summit, ID. (b) (left axis) Correlation between LIDAR snow depth and Retrieved total SWE on 03/19/21 using Sentinel-1 over Banner Summit for all points with mean temporal coherence greater than $\rho_{temp,threshold}$ versus $\rho_{temp,threshold}$. (right axis) Number of points in Banner Summit with mean temporal coherence greater than $\rho_{temp,threshold}$ versus $\rho_{temp,threshold}$. (c) Mean of Sentinel-1 temporal coherence between 12/01/20 to 03/19/21 over Mores Creek, ID. (d) (left axis) Correlation between LIDAR snow depth and Retrieved total SWE using Sentinel-1 over Mores Creek, for all points with mean temporal coherence greater than $\rho_{temp,threshold}$ versus $\rho_{temp,threshold}$. (right axis) Number of points in Mores Creek with mean temporal coherence greater than $\rho_{temp,threshold}$ versus $\rho_{temp,threshold}$.

total SWE for observation 15 with LIDAR snow depth is more appropriate. The correlation is high for observation 4 at the beginning of the snow season for both Banner Summit and Mores Creek. Observation 4 is after the first snow storm of the season. It shows that the spatial variability of snow at the end of the snow season is captured by the first or second snow storm. Although $\Delta SWE$ for station 21 is zero for the first observation, the correlation between LIDAR snow depth on 03/19/21 and total SWE on the first observation is relatively high, as seen in figure 13(b). As shown in figure 11(a), station 2 is in the

relatively low snow depth region. We believe there has been a snowstorm in the high-altitude region of Mores Creek. The high correlation is simply the correlation between LIDAR data on 03/19/21 and the first snowstorm in Mores Creek.

## 6 Conclusions

In this study, we used Sentinel-1 time series to retrieve $\Delta SWE$ and consequently total SWE. We chose a frame in Idaho that covers several SnowEx 2020-21 sites and 31 of SNOTEL in situ stations. LIDAR data are available for validating our results.

Sentinel-1 data was collected every 6 days over this SnowEx site instead of the regular 12 days which helps a lot with temporal coherence over snowstorms. This provides a unique dense time series of spaceborne data for studying the performance of SWE retrieval using InSAR.

  We showed that retrieved $\Delta SWE$ between two consecutive Sentinel-1 observations is highly correlated (0.8) with in situ values, with an RMSE of 0.93cm. For reference point of interferometric phase, we used two in situ stations with temporal

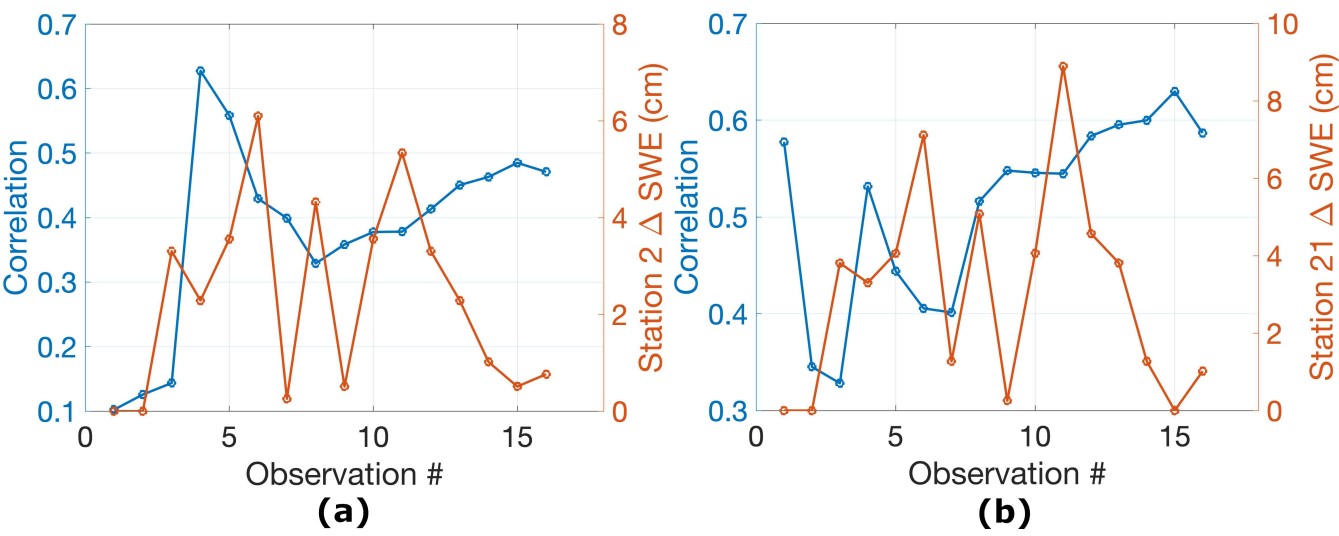

**Figure 13.** (a) (left axis) Correlation between LIDAR snow depth and Retrieved total SWE using Sentinel-1 on specific observation date over Banner Summit versus observation number. (right axis) $\Delta SWE$ (cm) for any specific observation date at station 2 in Banner Summit. (b) (left axis) Correlation between LIDAR snow depth and Retrieved total SWE using Sentinel-1 on specific observation date over Mores Creek versus observation number. (right axis) $\Delta SWE$ (cm) for any specific observation date at station 27 in Mores Creek.

coherence more than 0.35 and temperature less than $0°C$ for the entire time series. We subtracted the difference between the average of in situ and retrieved $\Delta SWE$ of these two stations from retrieved values to calibrate the retrieved $\Delta SWE$. The $\Delta SWE$ RMSE error is less than 2 cm for all stations and less than 1 cm for most stations. The correlation between retrieved and in situ $\Delta SWE$ is more than 0.6 for most stations. Interferograms with small average of in situ $\Delta SWE$ show low correlation between retrieved and in situ $\Delta SWE$. We demonstrated that low temporal coherence not only degrades the SWE retrieval performance, but also the unwrapping algorithm performance. We showed that big melting events between two Sentinel-1 acquisitions make the interferometric fringes noisy and unwrapping algorithm challenging. The retrieved total SWE has less than 2cm RMSE error compared with in situ values in the entire time series for 9 stations and more than 2 cm for 14 stations.

The highlight of the results of this study is the similarity between two independent measurements, retrieved SWE using Sentinel-1 data and LIDAR snow depth data. We used Sentinel-1 data between 12/01/20 to 03/19/21 to retrieve $\Delta SWE$ time series. By adding the entire time series of $\Delta SWE$, we calculated the total SWE on 03/19/21. Total retrieved SWE using Sentinel-1 interferometric data and LIDAR snow depth images over two regions in Idaho show similar patterns and are correlated by more than 0.47. We showed that the correlation is higher for regions with higher temporal coherence in Banner Summit.

Considering all these validations, we show for the first time that SWE retrieval using time series of InSAR *spaceborne* data is a very promising candidate for the future SWE mission.

We also showed that the main constraints for this method are temporal coherence, phase unwrapping, and phase ambiguity. We showed that snow storms reduce the temporal coherence significantly. Low temporal coherence reduces the accuracy of the interferometric phase and unwrapping algorithm. It is also shown in this study that melting due to warm temperature reduces the temporal coherence and the performance of unwrapping algorithm. Small SWE ambiguity at C-band (1.5cm to 3.5cm) makes the phase unwrapping more challenging. We think using in situ station as the reference point helps reducing the phase ambiguity error, at least locally, compared to other methods for referencing the interferometric images. If the temporal coherence is large enough for the entire image to reduce the phase unwrapping error, using the in situ SWE as the reference point reduces the phase ambiguity error in a larger region. Using a snow-free point or snow-free corner reflector as the reference point, cannot address the phase ambiguity in regions with deep snow. Going from C-band to lower frequencies such as L-band improves both the temporal coherence and SWE ambiguity. With the L-band NISAR launch coming next winter, the new dataset would be a great opportunity for global SWE retrieval.

# 7    Acknowledgments

The authors would like to thank JPL R&TD for providing funding for this project. The research was carried out at the Jet Propulsion Laboratory, California Institute of Technology, under a contract with the National Aeronautics and Space Administration (80NM0018D0004). The authors also would like to thank the reviewers for their very useful comments that improved sections 2.2, 4, and 5.2 significantly.

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
