# Peer review of "Snow Water Equivalent Retrieval Over Idaho, Part A: Using Sentinel-1 Repeat-Pass Interferometry"

_The Cryosphere, 2023_

## Referee Comment (RC1)

Gunneriussen (2001) $(cos\theta - \sqrt{\varepsilon(\rho) - sin^2(\theta)}$ and approximations divided by snow density $\rho$

Approximations vs. Gunneriussen (2001)

$\rho = 0.1...0.5 g/cm^3$, Gunneriussen (2001)

$\xi''(\theta, \rho, \alpha_{opt(\rho_{max} = 0.5)}/\rho$, Leinss (2015)

$\xi''(\theta, \rho, \alpha = 1)/\rho$, Leinss (2015)

$C(\theta, \rho)/\rho$, Oveisgharan (2023)

A from A * rho [+ B] ($\rho = 0.15...0.5$)

A from A * rho [+ B] ($\rho = 0.1...0.4$)

A from A * rho  ($\rho = 0.15...0.5$)

$\rho$=0.50

$\rho$=0.30

$\rho$=0.10

$\theta$ (deg)

---

## Author Comment (AC2)

We would like to thank the referee for their detailed and very useful review. We appreciate the reviewer's time and effort. We believe the manuscript is now in a much better shape. We hope this manuscript help the community in future approaches for SWE retrieval missions. We had two main changes to respond properly to the reviewer's concern.

- We used the line with zero interception for fitting.
- We used the average of two stations for calibrating the retrieved SWE.

We would like to thanks the reviewer for pointing to these two problems. The first one was needed and the second one make it easier for the readers to follow.

Please find our responses below.

**RC1, Anonymous Referee #1**

Review, Oveisgharan et al. "Snow Water Equivalent Retrieval over Idaho, Part A: Using Sentinel-1 Repeat-Pass Interferometry"

General comment:

The manuscript performs an analysis of a 6-day repeat pass InSAR time series with the aim to retrieve snow water equivalent (SWE) from the InSAR phase. The theoretical foundation of this SWE-retrieval method is known for more than 20 years but common InSAR problems like coherence loss, phase ambiguities, and the choice of the best reference value to correct for unknown atmospheric phase delays still pose significant obstacles for the method, in addition to the limitation to dry snow. The analysis of a time series with one of the shortest currently (locally) available repeat intervals of 6-days at a relatively low frequency (C-band) is a valuable contribution to assess the best combination of repeat-pass interval and frequency.

However, in its current form the manuscript has several shortcomings that needs to be addressed before consideration for publication:

1) The linear approximation of the equation from Guneriussen (2001) has a methodological shortcoming: a linear approximation (y = mx + c) is performed and after the approximation the y-axis intersection c is neglected. Correct would be to first assume the y-axis intersection is zero, and then estimate the slope m. Even though the estimated slope changes by several percent, it would not significantly impact the results of the paper; however the authors claim to provide a more general approximation compared to existing literature which is simply not correct; furthermore, the comparison of their estimates with literature (Fig. 10a) shows a significant deviation due to the methodological fitting mistake. In addition, their approximation method is not reproducible due to the lack of given ranges for snow density and incidence angles (see specific comments on section 2.2 and the attached figure).

- You are right. This is one of the two sections we changed significantly. We used c=0 and then did the fitting.

2) In section 5.1 the authors use the same data (average of all station's in-situ SWE measurements) for calibration of the InSAR based SWE estimate and for validation (local station in-situ SWE vs. local InSAR based SWE estimates). This means that the calibration data is not independent from the validation data (see specific comments on line 242-248/Figure 7a and b, 172a/b).

- I do not agree completely. We changed the average of all stations to average of 2 reliable stations and the results degrades a little. Even though we used the average of all stations to match the average of retrieved values, it's not similar to saying that we shift each station separately. This is something that has been done for soil moisture to remove the bias. The more the number of stations are the more the average is dependent to one or two of them or biased towards one of them. However, we understand that this will may confuse people and so we used two stations for calibration and the rest of them for validation.

3) In section 1 and 2, several statements are supported by reference to literature, however, in a very vague, not correct way or misleading way. These references should be improved (see specific comments below).

- We addressed all the ones you mentioned in the revised version of the paper.

4) In my opinion, low correlation coefficients (0.4...0.6) are often overrated by the authors as "very high/highly correlated/one of the best metrics". They need a more critical consideration.

- We changed the high correlated to correlated. However, w believe 0.5 is relatively good correlation. You mentioned 0.9 is a good correlation. I agree with that but the two datasets should be almost the same to have a correlation of 0.9. For now, I tried to tone it down.

5) Noteworthy, in section 5.2 the authors provide a reasonable correlation (0.51 and 0.62) between their spatially distributed SWE estimate and completely independent LiDAR data which supports the feasibility of SWE retrieval from 6-day repeat pass C-band data.

- Thanks! Yes, this was the highlight of this work.

Specific comments:

Abstract:

4: "optimal": Before specifying what the optimal parameters are, I would not use the work "optimal". I suggest "suitable"

- Done!

6: "long time series": In the sentence before, you talk about repeat intervals, therefore please specify if your time series has a 6-day or 12-day (or any more complex combination) repeat interval.

- We changed the sentence to: "In this study, we apply this technique to a long time series of 6-day temporal repeat Sentinel-1 data from the 2020-2021 winter."

10: "highly correlated with LIDAR": The correlation with the SWE station data (r = 0.82) is much stronger than with LIDAR data (r = 0.5). Therefore I would not say "highly correlated" which indicates that also a very good relation between snow height and SWE exists. However, dependent on the snow pack, density variations can deteriorate SWE estimations from snow height. Writing "highly correlated" here implies that both, radar interferometry and LIDAR estimates are equally precise which is physically not correct, because radar interferometry shows a physics-based, almost linear relation to SWE while LIDAR estimates require a guess (or auxiliary information) of the average snow density (as written in line 29). I suggest something like "well correlated".

- That's fair. We changed the highly correlated to well correlated.

32-38: I suggest a more critical review of the listed papers. The references are given in a context that suggests that SWE estimation with active sensors is generally possible from space. I think that is misleading and contradicts the abstract (line 3). Did all authors use spaceborne sensors and did they really estimate SWE? I suggest to better detail and quantify where the authors of the listed papers have been successful and where not - i.e., where are still current gaps that have not been filled by the listed authors?

- We rewrote this paragraph trying to explain the limitation of each. The limitations for the ones with the interferometry technique are explained later in next section. "Active microwave sensors provide high resolution and global coverage. There has been many efforts in the last two decades trying to estimate SWE or snow depth using active sensors mounted on a tower (Cui et al., 2016; Lemmetyinen et al., 2018; 35 Ruiz et al., 2022; Leinss et al., 2015), airborne (Marshall et al., 2021; Nagler et al., 2022), or spaceborne systems (Lievens et al., 2019; Liu et al., 2017; Conde et al., 2019; Dagurova et al., 2020; Eppler et al., 2022). Backscattered power from active sensors is used to estimate SWE (Rott et al., 2010; Ulaby and Stiles, 1980; Cui et al., 2016; Nghiem and Tsai, 2001; Lievens et al., 2019). A dual-band (X and Ku) SAR mission has been the focus of the European Space Agency (ESA) and Canadian Space Agency (CSA) for SWE spaceborne measurements (Rott et al., 2010; Lemmetyinen et al., 2018). However, accurate *a priori* characterization of snow micro-structural parameters is of primary importance in the accuracy of SWE retrieval algorithms using backscattered power (Lemmetyinen et al., 2018; Durand and Liu, 2012; Cui et al., 2016). The most common *a priori* characterization used for SWE retrieval algorithms using backscattered power is grain radius. This has been done using passive data; however, the methods are restricted to passive retrieval errors and also mismatch between active and passive resolutions. The ratio of cross-pol to co-pol Sentinel-1 backscattered power has been used to estimate snow depth over mountainous regions with deep snow (Lievens et al., 2019, 2022). Using Sentinel-1 backscattered power ratio is a unique approach showing the success of snow depth retrieval using the spaceborne radar time series data. However, the retrieval mostly works for deep snow in mountainous regions. The radiative transfer physics at C-band for this method are still poorly understood.

47-49: These lines imply that FMCW radars cannot be used in space because of their wide bandwidth. That is not correct. FMCW radars are cheaper because they do not require temporal pulse compression (that needs high-power electronics) but because they transmit and receive at the same time, they are limited to a range of a few km because the signal travel time limits the pulse repetition frequency. And it would even be simpler to build them with a narrow bandwidth, but a benefit of these experimental and locally installed systems is the possibility to explore wide bandwidths. It is more, that wide-band system cannot be used in space because of limited frequency allocation. Possibly, I misunderstood your point: Did you mean nadir-looking radar altimeters (that could be FMCW from air or pulse-systems from space). Please clarify.

- FMCW systems needs to have wide bandwidth in space to avoid ambiguity. Something that we do with prf for pulsed signal. We should be able to estimate the distance from the received frequency. With that wide bandwidth for large distances we have the problem of bandwidth allocation for active sensors. We add this clarification to the paper: "These sensors need a wide bandwidth for spaceborne large distances. Due to allowable frequency bandwidth of a spaceborne active sensor (Tao et al 2019), FMCW system cannot be used in spaceborne missions for global coverage due to their wide bandwidth."

50: "specularly reflected": that is not correct in the context of the given references. "specularly reflected" means "like on a mirror", according to the law of reflection. However, the listed authors observe the snow/ground surface from the same point as the illumination source (monostatic or almost monostatic radar system).

- The authors wanted to show the theory behind SoOp (specular reflected signal) is very similar to differential interferometry. But we realized it hasn't been communicated correctly in the paper. Now it is rewritten: "The phase change of specularly reflected signals in Signals of Opportunity (SoOp) is shown to be strongly dependent on SWE changes for dry snow and on depth changes for wet snow (Yueh et al., 2017, 2021; Shah et al., 2017). The phase sensitivity to SWE changes increases at higher frequencies. However, the temporal coherence decreases and the phase ambiguity increases which makes the phase unwrapping very challenging at higher frequencies. The theory behind this method is similar to repeat pass interferometry that is explained in section 2. The advantage of this method is that the stratigraphy of the snow has little impact on the SWE retrieval (Yueh et al., 2017) similar to SWE retrieval explained in section 2. Using the long wavelength signal at P-band in SoOp is very helpful for addressing the loss of temporal coherence and phase unwrapping challenges of this method. However, there has been very limited data showing the success of this method at P-band. Achieving high resolution is another challenge of this method."

52: "in wet snow the phase center is normally at the snow surface": that might be true for X-band and above, but the cited papers used P-band where I would expect a significant penetration into the wet snow pack. Did the cited authors really show that the phase center is at the surface? Please clarify or correct.

- It has been shown in section III.E of Yueh et al., 2017, that snow wetness brings the reflection from the ground-snow to air-snow. They also show that the phase change is

proportional to snow depth for wet snow. You are right that the signal should penetrate to some extent into the snow at P-band but the main signal will come from the snow ground interface for specular reflection. For backscattered differential interferometry geometry (our retrieval method in this paper) the phase may not be that linearly dependent to snow depth change.

53: "the phase unwrapping (...) increases at higher frequency." That does not make sense. Either "the effort for successful phase unwrapping increases at higher frequencies" or "the phase wraps more frequently at higher frequencies."

- That is correct, thanks for pointing that out. We changed it to: "The phase sensitivity to SWE changes increases at higher frequencies. However, the temporal coherence decreases and the phase ambiguity increases which makes the phase unwrapping very challenging at higher frequencies."

55: "the theory behind this method": Could you first concisely describe the method you are referring to? What's the core idea of the method (see Guneriussen 2001)? After that, you can outline successful applications and limitations of the method as attempted in line 50-54.

- We think with the new write up of this paragraph (written in response to 50), it is clearer.

56: snow stratigraphy and grain size: you are citing Yueh(2017) here, but consider also citing Leinss(2015) who did an error analysis on the impact on SWE estimates from layers with different density.

- Good point, that one is more complete, thanks for pointing that. Done!

80: Please provide a reference after "dual-pol dual-frequency retrieval algorithm". Are you sure that this algorithm relies on the assumption of non-scattering, non-absorbing dry snow, so that microwaves penetrate to the ground?

- The reference is added. We also add a sentence to better explain why we need the wave to penetrate all the way to the ground. We need to make sure that the phase change is due to phase delay from the ground not phase center shifted to somewhere in the snow volume. Here is our modification "Similar to the dual-pol. dual-freq. retrieval algorithm (Lemmetyinen et al., 2018; Cui et al., 2016), this technique relies on the dryness of snow in order to penetrate all the way to the ground and the scattering from the snow layers and snow volume is minimized compared to snow-ground return.

90: "highly correlated": Please quantify by providing a correlation coefficient.

- Done by adding it to the text: "The correlation of 0.76 was observed between the retrieved SWE change using L-band UAVSAR differential interferometry between 2/1/2020 and 2/13/2020 and the collected LIDAR snow depth change between 2/1/2020 and 2/12/2020 over the open regions of Grand Mesa in dry snow conditions (Marshall et al., 2021)."

95: "works well": Could briefly you point out how you solved the problem of phase reference and how you corrected for atmosphere?

- In section 4 we explained the details of our retrieval. Here we are just explaining how our approach is different from others as our retrieval covers a long time series with many in situ stations for validation. Describing the detail of reference point and atmospheric removal, even briefly, seems out of context here. However, for your reference, we used mintpy to remove atmospheric noise and we used the average of reliable in situ stations as the reference point.

106: "a high temporal coherence is observed...with a month temporal baseline": Could you provide some quantification or statistics from the cited paper? Was this "high coherence" observed in a single observation or was it generally high over a certain time series? Under which snow conditions?

- We changed it to "A medium mean temporal coherence of 0.41 is observed at L-band between two winter seasons in shrub-lands with 10.2cm average snow depth (Molan et al., 2018)" to make it clearer. The paper is not focused on snow in particular but has used the interferograms over snow and snow is one parameter in their modeling.
- Figure 2(c) of the cited paper shows coherence between 0.4 to 1 for one month observation in shrubland but because the snow depth change isn't mentioned we decided to not use that and talk about the average between two winter season.

109: What do you mean with "controlled system"? What snow conditions did the authors observe?

- We removed the controlled system as we thought it may be confusing for the reader. What we meant was that they were able to distinguish the vegetation was almost frozen in the snow and doesn't move, they distinguish one or two occasions that there was windy nights, … . However, in a real big scene we cannot distinguish these scenarios. In addition they mentioned "All acquisitions were coregistered to compensate range-shifts due to a snow related signal delay" which needs some information from the scene for that I believe.

110: "the ... temporal coherence... Leinss (2014)": The cited paper does not deal with temporal coherence. Please check. I guess, you mean Leinss(2015).

- That is correct. We fixed it.

114: "the temperature was shown to be the most critical variable ...": In which sense? There are strong differences between positive and negative temperatures. Variations in negative temperatures should not significantly affect the coherence.

- We added this sentence to the paper "Temperature above zero reduced the temporal coherence drastically". Although as the temperature decreases below zero the coherence increases gradually in most cases and different frequencies in the referred paper.

120: I would drop the complete sentence "More studies are needed..." because it is very general.

- Agreed and Deleted!

Section 2.2:

Comment for section 2.2: The described method seems to contain a few methodological shortcomings, misses values of parameters that are needed for reproducibility and is, in my opinion, less general than the approximation of Leinss (2015). Even though I do not expect any major impact on the results, I would like to urge the authors to improve this section and to set it in proper context with existing literature, specifically Gunneriussen (2001) and Leinss (2015) but also Mätzler (1996) and Wiesmann (1999) [reference below, comment on l.128]. An interesting idea of the described method is to fit a function for every incidence angle - this idea is, however, almost identical to the introduction of the fit parameter \alpha in Leinss (2015) to provide an optimal numeric approximation for every chosen incidence angle, because the incidence angle is very often precisely known - in contrast to the snow density, for which only a reasonable range can be estimated. In contrast to Leinss (2015), where a density range from 0 to a maximum snow density (rho_max) was assumed, a slightly improved approximation could possibly be achieved by assuming/selecting/limiting snow density to a more realistic range, e.g. rho = 0.1...0.5 g/cm^3, or whatever the authors expect for their specific region.

- As you can see below, we changed the section to fit a line to it. Thanks for pointing this out. However, we still think using one equation for the entire incidence angle range is very convenient. So, by this analysis we can see what the error is using this approximation. However, we can use different fitting for each angle too. However, it makes things much faster for a big scene to use just one formula.

Specific comments for section 2.2:

125: "related the interferometric phase directly": Could you be more specific? I suggest "related the interferometric phase with a linear approximation directly to SWE changes"

- We changed it to "With some approximation to equation 1, Leinss et al. showed a linear relationship between the interferometric phase and SWE change (Leinss et al., 2015)."

126: "depends on the range of incidence angles and snow density": That is not exactly correct. I suggest:  "depends on the chosen incidence angles and the maximum expected snow density"

- Fair enough, we changed it to: "The approximation depends on the incidence angle of each pixel and the maximum expected snow density."

126: Could you specify what exactly you mean with "more generalized"? In my opinion, your method is less general then the approximation by Leinss (2015), because it is limited to snow densities up to 0.5 g/cm^3 and incidence angles between 20 and 50 degree.

- I think the fact that we have "one" formula for any point in the image to convert the phase to SWE makes it generalized. You are right that we know the incidence angle and we use the corresponding formula for that incidence angle but I think it makes it more time consuming. The density is limited to reasonable snow densities observed and incidence angle is what we generally observe in Sentinel-1 images. So, in applying the formula to a SAR image in Snow it is generalized. Although the incidence angle can get larger as can be seen in figure 1(c) and the error remains less than 10%.

127: What are the units that you use for density? kg/m^3 or g/cm^3 or volume fraction (unitless)?

- It is g/cm^3 as explained in the figure but added here too.

128: "We use Matzler's model for calculating epsilon in equation 1 (Mätzler, 1987).": Could you provide a more specific reference (e.g. equation number or show the used equation for epsilon)? The cited dissertation has 130 pages and contains many different models for the permittivity of snow. Note, that there are more recent equations for the permittivity of snow from Mätzler, e.g.:

- We added the corresponding equations in the text.

  - C. Mätzler, "Microwave permittivity of dry snow," IEEE Trans. Geosci. Remote Sens., vol. 34, no. 2, pp. 573–581, 1996-03, doi: https://doi.org/10.1109/36.485133.

  - A. Wiesmann and C. Mätzler, "Microwave Emission Model of Layered Snowpacks," Remote Sens. Environ., vol. 70, no. 3, pp. 307–316, 1999, doi: http://dx.doi.org/10.1016/S0034-4257(99)00046-2 (Note, that Eq. 46 misses an exponent of 1/3 for eps_s which becomes apparent when comparing with Eq. 10, Sect. IV-F in Mätzler 1996).

129: Please specify what units you use for SWE: kg/m^2, mm w.e.q., m^3/m^2=m (w.e.q)? Make sure, that the equation returns the correct values when entering SWE, rho, k_i, and C in the specified units.

- We double checked all the unites and add a sentence to be clearer. "Note that the $\varepsilon$ and consequently C are unitless, $\rho_{water} = 1$ g/cm$^3$, $\Delta$SWE\$ is in (m), and $\kappa_i$ is in (1/m)."

Figure 1a: What units has C(theta, rho)? Could you verify if you really plotted C(theta, rho) as defined in Eq. (2). When I compute C from equation 2, I obtain the data shown in Fig. 1a, but only when not dividing by snow density rho. That means, currently, Fig. 1a is identical to Figure 8left, in Leinss (2015), and C(theta, rho) == xi(theta, rho_s); an overlay of both figures confirms this. I think the division by rho in the definition of C should be removed.

- You are right, thanks for noticing that! I didn't notice this is the same as figure 8 in Leinss paper. I moved the rho to equation. However, in terms of simplification of equations and plots, the rest are correct.

**Note:**

For the rest of the comments for this section, I completely agree with you. Thanks for your great suggestion! I think I was fitting and then notice B is very close to zero, … . But as you mentioned it makes more sense to assume B=0 from the beginning. I rewrote this section assuming B=0. I also specified the ranges for incidence angle and density. I agree that depending on that range the "A" changes. Although it doesn't change the results significantly. So, I use incidence angle range of Sentinel-1 data over this frame (0 to 80) and snow densities as mentioned before. Using the new approximation I redid all retrievals in the results section. The results for in situ stations remain almost the same and the correlation in lidar section degrades very little as expected.

132: "we fit a line to C for different incidence angles": Could you specify the interval of densities rho that was used for fitting? Depending on the chosen interval, the fitting parameters can vary by several percent (see attached figure).

- Rewrote the section.

132: Could you specify the interval of theta for that lines-slopes were obtained by fitting? In the attached figure I assumed 20..50°, similar to Figure 1b and 1c.

- Rewrote the section.

135: I cannot reproduce the equation in line 135. When following the described approach (assuming a density range of rho = 0.15..0.5 g/cm^3) my values for the parameter A deviate by 1-2% from eq. 135. (see attached figure, dashed red line vs. solid blue line). Also note that for a different density interval of rho = 0.1..0.4, the values of A deviate 3-4% from eq. 135 (dashed purple line).

- Rewrote the section.

136: "Assuming B(theta) = 0": I see this as the main shortcoming in this section: Why do you first fit a line including an y-axis intersection, and then assume that the y-axis intersection is zero? As shown in the attached figure, you obtain more accurate results when first assuming "B(theta) = 0" and then fit a line C = A(theta)*rho where the y-axis intersection is zero by definition (solid teal line). The difference to eq. 135 is about 5% at incidence angles of 20°.

- Rewrote the section.

136: "B is very close to zero": (See also comment above). How close? Neglecting terms makes only sense when putting them into relation to larger terms. Could you put B in relation to A * rho? The ratio of B/A is up to 1.5%, depending on the chosen fitting interval; however, you need to consider the equation C = A(theta)*rho + B(theta) where the snow density (assuming units of g(cm^3)) can be very small, again, depending on the chosen interval. With rho = 0.15, B is as large as 16% of (A*rho) which can have - and has - a significant impact on the fitting results.

- Rewrote the section.

139: What's the advantage of using eq. (4) vs. Eq. (18) from Leinss 2015? Considering the above described shortcomings, I do not see any advantage of using this equation. But a proper linear fit as suggested two comments above would do it.

- I think our equation is applicable to wider range of snow densities and incidence angles which makes it convenient for Sentinel-1 frame.

Section 3.1:

143-151a: This paragraph reads than a general description about Sentinel-1 and the ASF. Could you describe which data and which processing workflow you use for your input data?

- We edited to better show what we used for our workflow: "We used the Interferometric Wide Swath (IW) mode data with 5 and 20m single look resolution in range and azimuth direction, respectively. The IW swath width is about 250km. We used ASF On Demand Processing to generate interferometric phase and coherence at vv and vh (transmit-received polarization) polarization. Alaska Satellite Facility's Hybrid Pluggable Processing Pipeline (HyP3) is a service for processing Synthetic Aperture Radar (SAR) imagery. The workflow includes interferometric phase correction for ground topography and geolocation. The ASF HYP3 uses a Minimum Cost Flow (MCF) algorithm for phase unwrapping. The unwrapped phase and interferometric correlation were used in this study."

143-151b: Which incidence angle (range) did the acquisitions have that you processed?

- The incidence angle for the frame we processed varies between 0 to 80.

152-172: same as above: I would expect an introducing sentence to motivate why you discuss the SnowEx2021 campaign(s) and the SNOTEL data. The reason why these campaigns are discussed is given at the end of the section, line 171-172. It would be easier to read if these sentence appear as a motivation/reason at the beginning of the paragraph. (see also next comment).

- I added this sentence right before the SnowEx campaign as an introduction: "Sentinel-1 collects data every 12 days globally but has the capability to collect the data every 6 days over targeted areas, mainly over Europe and selected areas such as SnowEx sites. In order to validate our SWE retrieval using Sentinel-1 data, we use LIDAR data from SnowEx campaign and SNOTEL data as discussed in section 5. We also use the average of SNOTEL data as a reference point for SWE retrieval, as seen in section 4."

162/Figure 3b: Could you add to the caption in Figure 3b that for Delta SWE the first date of the differenced dates is shown?

- Done. We add this sentence to 3(b):" Note that the Delta SWE is marked on the first day of each observation.

163: "located in remote, high elevated mountainous regions": Figure 3a shows strongly variable SWE values. Is that due to altitude differences of the stations? Could you specify the altitude range of the stations?

- This is nice feature of using Spaceborne data. It covers a wide range of stations with different SWE values. And using the average of SWE change of all those stations doesn't necessarily bias the result. We add this sentence to the text:" Different colors show different SNOTEL stations. The elevation of these stations varies between 3200m to 9520m. Therefore, the large spread of SWE between different stations in figure 3(a) is expected."

170: "As seen in this figure [3a], there is a one 6-day repeat data acquisition gap in Sentinel-1 data on 2/5/21": Figure 3a shows two gaps as if there would be no acquisition on 2021-01-30 which is not true (Figure 4/5 show the pair 2021-01-24 vs. 2021-01-30).

- As mentioned in the paper, the dashed lines show the first day of each interferograms. There was observation on 01/30 but hasn't been shown here as there is no 6 days observation that

starts on 01/30. We understand that it is a little confusing but we want to identify the dates as appear in figure 3(b).

172a: "we used these in situ data for SWE retrievals performance": what do you mean? To tweak the performance? Or to validate the performance?

- Changed it to: "SWE retrieval validation".

172b: [used for] "the InSAR reference points": Could you clarify which data you used for InSAR phase reference points and which data for validation of InSAR-based SWE results?

- We add this sentence for more clarity: "We used the SWE data from these in situ stations for (a) SWE retrievals validation by comparing retrieved ΔSWE with SNOTEL ΔSWE(as seen in section 5.1), and (b) the InSAR reference point by subtracting the average of two SNOTEL ΔSWE from the retrieved ΔSWE (as explained in section 4)."

174-175: "LIDAR...are reliable sources of validation data, particularly a powerful constraint for InSAR retrieval of SWE": same as comment above: Could you clarify if you used the data for validation or to constrain the results?

- We added the sentence:" We used the LIDAR data for validating the retrieved SWE results."

175: "The "SnowEx20-21 QSI LIDAR DEM 0.5m" data set is part of the SnowEx 2020 and SnowEx 2021 campaigns (Adebisi et al., 2022). The data includes digital elevation models, snow depth, and vegetation height with 5m spatial resolution." - Could you clarify which data had 0.5m resolution and which 5.0 meters?

- Thanks for noticing that, it was a typo. We changed the 5m to 0.5m.

185: Could you explain why you selected the period from 2020-12-01 to 2021-03-30? Did you limit the period to dry snow conditions?

- We added this sentence to clarify: "We selected this period to (a) capture most of the seasonal snow storm and (b) avoid wet snow as much as possible."

185-186: Do you have any information about (or did any correction for) the height-(or air pressure) dependent phase contribution? It seems yes, maybe add a reference to Section 4.1.

- I am not sure what you mean here. We use pyapps to remove troposphere noise which is dependent on the topography, and we mentioned that reference in section 4.1. So, I am not sure what you are requesting here.

189/191: "In this study, any pixel with less than 0.35 temporal coherence  is not considered reliable (...) However, in .. 5.1.2 and 5.2 we used all ... data even with low coherence to calculate total SWE": Why do you consider it as an advantage to use all data, even though you consider the data as unreliable due to low coherence? Or is there something specific about these two sections that the

reader has not yet been informed yet? Would it not be more of advantage to set unreliable data to some assumed SWE change, e.g. zero? (see also comment on line 197)

- We changed it for total SWE calculation and now we just use the data that is reliable. However for the LIDAR data, we used all the data because we are calculating total SWE and if we miss on day i it will affect the total SWE on day i+n. So, if we have a lot of SWE on day i, we are $\Delta SWE_i$ less in the coming days, hence divergence. So, we consider all the data assuming there is some error in that. We saw that even with that assumption, the results are still good enough. If we consider zero for SWE on day i, our results are $\Delta SWE_i$ away from the actual total SWE at the end. We added this sentence to make it clearer: "The reason is that in order to compare the total SWE at each day, we need the whole $\Delta SWE$ time series up to that date."

192: "The ionospheric error ... is much smaller than other sources of error .. and considered negligible": Could you distinguish tropospheric error and ionospheric error? Or is the ionospheric error removed together with tropospheric phase delays?

- Ionospheric error is due to ionosphere delay and is negligible at C-band. It mostly lies between 85km to 600km above earth surface. The atmosphere is ionized and contains plasma in this region. The total electron content is what affects the phase delay of InSAR. It is dispersive for interferometric phase which means that it is proportional to square of wavelength. Therefore, the effect at L-band is much bigger. We ignore its effect on C-band Sentinel-1 data. Troposphere is mostly up to 12km above earth and is nondispersive meaning that it doesn't depend on the radio frequency wavelength. We remove it by models as explained in section 4.1. We added this sentence for more clarification: "The radar signal propagating through ionosphere is delayed. The delay is a function of frequency of the signal, Earth's magnetic field, and total electron content (TEC) and affects the accuracy of $\Delta SWE$ retrieval. The ionospheric error at C-band is much smaller than other sources of error and we consider it negligible in this study."

197: "similar to correlation filtering, for ... 5.1.2 and 5.2 we used all the ... data": why? (see comment 189/191)

- We changed it for section 5.1.2 and just compared the total for reliable SWE but still kept all the data for LIDAR comparison. We understand that it will affect our accuracy but there is no other workaround solution. We add this sentence: "Similar to correlation filtering, for the results in section 5.2, we used all the time series data, even with temperature more than zero. Similar to temporal coherence, the reason is that in order to compare the total SWE with LIDAR snow depth at LIDAR acquisition date, we need the entire $\Delta SWE$ time series up to that date."

201-203: For choosing the phase reference point, Tarricone et al. (2023) point out a promising idea for mountainous terrain: They suggest snow free areas, derived from optical data (fractional snow cover maps), as phase reference. I think it's worth to cite this paper here.

 - Tarricone, J., Webb, R. W., Marshall, H.-P., Nolin, A. W., and Meyer, F. J.: Estimating snow accumulation and ablation with L-band interferometric synthetic aperture radar (InSAR), The Cryosphere, 17, 1997–2019, https://doi.org/10.5194/tc-17-1997-2023, 2023.

- Thanks for pointing that out. We add it to the paper: "For ΔSWE estimation using InSAR, the reference point is chosen either by corner reflectors (cleaned of snow) with stable zero phase (Nagler et al., 2022; Dagurova et al., 2020) or using the average of in situ ΔSWE (Conde et al., 2019) or using a snow free region (Tarricone et al., 2023)"

**Note:**

We changed the reference point from average of all SWE values to average of two in situ stations with good temporal coherence and temperature less than zero for the entire time series. This way we avoid the same calibration and validation problem. However, for correlation coefficient calculation we need to remove the mean of the two variables, by definition. So, I think even using the average of all stations shouldn't matter much. Note that our calibration doesn't involve any model fitting. It is just the subtraction of a number from the entire image. The subtraction of mean of in situ station is being used for soil moisture unbiased estimation. In any case, I think using the average of just two stations would address most of your comments for this section. The results degrade a little bit but not significantly. Below please find the responses to each one.

204-206. "we used the average of all in situ Delta SWE ... to calibrate the retrieved [InSAR] Delta SWE images": Figure 4 (and also Figure 3) shows strongly variable SWE values across all stations. Why would a correction with the average improve the estimates? Would a correction using stations with small/no SWE change (I would assume, these are the stations at lower altitude) not be more beneficial (see also Tarricone TC 2023)?

- Normally the places with low snow change and as you mentioned low altitude go through melting more regularly. The melting and temperature are one of the main sources of error in this method. Even for the areas with no snow change soil moisture affects the phase and consequently SWE change(you can see that in phase closure studies in recent years). On the other hand, in the areas with lots of SWE change, phase unwrapping and phase ambiguity are a challenge with low temporal coherence. So, average of all may not be the best option but it is one way to go. However, as mentioned we used two stable in situ stations now for calibration and this is not a problem anymore.

210-211: Could you provide the given information (Delta SWE, and the information that there was no SWE change and two storms) (also) in the caption of Figure 4?

- Done! We add this sentence to the caption: "The average of in situ ΔSWE, for images (a), (b), and (c) are 0.01cm, 2.72 cm, and 4.33 cm, respectively."

242/244: "all the retrieved Delta SWE" / "all the retrieved Delta SWE time series": It seems you plotted Delta SWE between each two S1 acquisition for all SNOTEL stations. Even though you analyzed data from the whole time series of images, the shown data does not contain any specific time-series characteristics (in contrast to section 5.1.2). Is that correct? If so, clearly indicate this.

- That is correct. We add this sentence for more clarity: "Note that the data shown in figure 6 is the SWE *change* between two consecutive Sentinel-1 data that are 6 days apart. We showed the ΔSWE for all stations and all consecutive observations between 12/1/20 and 3/30/21."

242-248/Figure 7a and b: What is the correlation coefficient between the mean Delta SWE values of all stations (that was used as reference for the Delta SWE images - see line 204-206) vs. the Delta SWE of all individual stations? - The contrast of the good correlation between 0.6 and 1.0 (with a mean around 0.82) in Figure 7b (and the mean in Fig. 7a) and the low correlation (between at least -0.5 and +0.8, median correlation at approximately 0.4) is a hint that the major part of the correlations is due to the fact that validation data was used for calibration.

For estimating a lower limit for the expected error in your data: how large is the standard deviation for certainly snow free areas (if there are any)?

- We changed the calibration method and used just two stations. So, hopefully this resolves the issue here. I didn't get what correlation coefficient you are after. As mentioned above, snow free regions may bring soil moisture information in SWE retrieval. So, that cannot be a good choice for calibration nor for minimum SWE error calculation.

252: I would not call an correlation coefficient of 0.4 and higher "very good" (See examples on https://en.wikipedia.org/wiki/Pearson_correlation_coefficient). I would call a correlation of 0.8 good, and a correlation of 0.4 poor.

- We changed very good to good. 0.8 is considered highly correlated. 0.4 you can definitely see there is a relationship.

Figure 7c: What is the correlation of the first acquisition at 12/01? Could you scale the y-axis so that all datapoints are shown?

- The SWE change between day 1 and 2 is zero. Therefore, the correlation coefficient is NaN.

Figure 7c: Could you add to the caption in Figure 7c that for correlation and the first date of the correlated date pair is shown?

- We added this sentence to the caption: "Note that the labels on x-axis show the first date of each interferometric observation."

250-262: Here, you discuss reasons why you could expect a low correlation. If you select only acquisitions pairs with SWE changes (average over all station) larger than a certain threshold (according to Figure 3a) how would the correlation coefficient look like? For example, you could have the average SWE change per acquisition pair (sorted by magnitude) on the x-axis and the correlation coefficient of all SNOTEL stations vs. InSAR Delta SWE on the y-axis, would the correlation show an increasing trend with increasing SWE change? I would expect that if there's a clear correlation between Delta SWE and the station data. In contrast to Figure 7a, such a plot would be independent on the choice of the reference phase (or reference Delta SWE).

- The argument that when ΔSWE is small our retrieval doesn't have enough accuracy and therefore we won't expect to get a good correlation between retrieved and in situ is different from saying that ΔSWE is correlated to the correlation between in situ and retrieved. Having said that I plotted as you suggested for my curiosity and as you can see below there is some dependency. However, there are just 18 data points, and we cannot conclude anything. In any case, I think my point here is that for days that we have very small SWE change, it is

expected that we don't see good correlation between in situ and retrieved SWE as the noise is bigger than SWE change.

[Figure]

Section 5.1.2: Note, that the methodology in this section might suffer from the same calibration problem as commented on 242-248. (validation data used for calibration). If you plot the SWE error at the end of the season (03/31) over total SWE, I would expect that for stations with a large total SWE the S1 data show a negative bias, while for stations with a small total SWE, S1 shows a positive bias.

- I believe with the new reference point currently defined; this issue is resolved.

277/281: "we think the main reason for divergence is the ... phase ambiguity" vs. "The divergence is mainly due to phase ambiguity": These two statements are not compatible with each other. Do you assume/think that that's the main reason? - then you can't make the second statement. Or do you have sufficient evidence that that's the reason? - than provide the evidence it, instead of "thinking" it's the reason.

- We changed the " the divergence is mainly due to phase ambiguity" to "We will investigate the reason behind the divergence of retrieved SWE from in situ SWE of these stations in the future work of this study."

Figure 10a: I think the main reason for the discrepancy between the approximation from Oveisgharan (2023) and Leinss (2015) is the shortcoming of correct approximation of the equation from Guneriussen (2001). See attached figure (blue line vs. thin black line; even more accurate: blue line vs. thick black line). See comment on line 136.

- You were right. As mentioned above, I used a line going through zero and the result is very close. Thanks for noticing that.

304-309 vs. 314-318: These two paragraphs show a very high redundancy of text. Consider writing it more concise, e.g. first a single paragraph describing both figures (11 and 12), then a second paragraph describing the differences.

- Done as requested, good suggestion, thanks!

308-309: "The high correlation of 0.51 shows the success of this method": I would say "there's likely some correlation supporting that SWE estimation with InSAR from space is feasible."

- The patterns are very similar in the image and that amount correlation is not small. We changed the writing to: "The relatively high correlation (0.47 and 0.59) between the two independent measurements with different resolutions is a very good indication of the success of this method in estimating SWE."

318-319 "The correlation ...is 0.62. This high correlation is one of the best validation metric for SWE retrieval using the InSAR techniques". First, I would not consider a correlation of 0.62 has high. It basically shows that there is likely some correlation. Considering that the comparison of the spatial snow height distribution from LIDAR is a completely independent dataset from the spatial distribution of the InSAR estimate (that includes calibration by in-situ data from SNOTEL station) makes this correlation, more correctly, the two correlations shown in Figure 11c and 12c, currently the most convincing correlations in this manuscript.

- We agree that this result is the most convincing result we showed in this work. Having said that the other ones are quite impressive. As mentioned for the previous comment we changed the writing to better address your concern: "The relatively high correlation (0.47 and 0.59) between the two independent measurements with different resolutions is a very good indication of the success of this method in estimating SWE."

326: "highly correlated (0.82) with in situ values" - as commented above, there might be a calibration problem as commented on 242-248. (validation data used for calibration).

- With the new calibration method, that problem doesn't exist anymore.

327: "The retrieved total SWE has less than 2 cm RMSE compared with in situ values in 16 stations" - you neglect here, that it is worse for the other 34 stations of the in total 50 stations.

- Not all of those stations are in our image. There are 43 stations in the rectangle shown in figure 4 but 12 of them are in the blue region where there is no data there. Therefore, we have 31 stations inside Sentinel-1 frame. I changes figure 6(b) to only show the 31 stations and not the ones that are out of frame. As added now to the paper: "Among all 31 stations in the Sentinel-1 frame, 6 of them have temporal coherence less than 0.35 or temperature more than zero in their entire time series. Two of them are used for calibration of the phase. So, there were 23 stations with more than 2 reliable observation dates in their time series. Among the 23 stations, 9 have SWE error less than 2cm (green diamonds) and 14 of them have SWE error larger than 2cm (red diamonds)." So it is actually not that bad.

328-329: "We show ... that SWE retrieval using spaceborne InSAR timeseries is a very promising candidate for future SWE missions": It might be true that the analysis of InSAR time series is a promising candidate for future SWE missions. However, your paper shows that the method is likely feasible, but it also shows that several problems of the method, like phase unwrapping, loss of coherence, choice of reference phase, are still not sufficiently solved and that even with a 6 day repeat interval at C-band it is not easy to retrieve reliable SWE estimates.

- That is true and we also mentioned that in the next paragraph. "We also showed that the main constraints for this method are its temporal coherence, phase unwrapping, and phase ambiguity. We showed that snow storms reduce the temporal coherence significantly. Low temporal coherence reduces the accuracy of the interferometric phase and unwrapping algorithm. Small SWE ambiguity at C-band makes the phase unwrapping more challenging. Going from C-band to lower frequencies such as L-band improves both the temporal coherence and SWE ambiguity. With the L-band NASA-ISRO SAR mission (NISAR) launch coming next winter, the new dataset would be a great dataset for global SWE retrieval." I added "we showed" to this paragraph to confirm that we showed these problems in this study too.

I'm looking very much forward to the first SWE time series from NISAR :)

- Yes, let's go NISAR!

Technical comments:

Please check the paper against the submission guidelines https://www.the-cryosphere.net/submission.html, specifically dates should be written as "dd month yyyy" (https://www.the-cryosphere.net/submission.html#math) rather than in American notation. I know that sometimes (e.g. in figures or tables) the date can be given according to the international ISO 8601 standard (https://en.wikipedia.org/wiki/ISO_8601) but that needs to be confirmed with the editors.

189: You might want to change double negation to a positive formulation "less than .. not considered reliable" -> "higher than ... considered reliable"

- Changed it to :" In this study, any pixel with temporal coherence more than 0.35 is considered reliable."

Figure 4 and 5: It might make sense to combine both figures into a six-panel image with a top row of the three Delta SWE images and a bottom row with the three correlation images.

- Done!

---

## Author Comment (AC4)

We would like to thank Jorge Jorge Ruiz for his very useful review. We appreciate the reviewer's time and effort. We believe the manuscript is now in a much better shape. We hope this manuscript help the community in future approaches for SWE retrieval missions. We would like to point out that we have changed the reference point from the average of all in situ stations to the average of two stations SWE change with reliable time series. This will address your concern about reference point.

Please find our responses below.

**RC1, Referee #2, Jorge Jorge Ruiz:**

The manuscript from Oveisgharan et al presents the results of the analysis of Sentinel1, C-band, InSAR timeseries over Idaho for Snow Water Equivalent (SWE). The generation of the InSAR derived SWE timeseries over the SNOTEL stations was demonstrated using them as reference for phase. Additionally, the total SWE retrieved over several months showed great agreement with SWE when compared to LIDAR snow depth. The work is of interest for the snow community focusing on InSAR SWE retrievals. The results presented in this manuscript would be useful for future satellite missions such as NISAR or ROSE-L. Furthermore, it also demonstrates the technique using Sentinel1, that will operate along ROSE-L. Congratulations to the authors for the great work. I have some concerns I think should be addressed before publication.

- Thanks for your useful review and compliment! Your feedbacks improved the paper a lot.

General Comments:

1. I think an interesting addition would be a figure (maybe in the annex) with the time series of in situ vs retrieved SWE for all stations. This is at your consideration. Also, in Figure 2.b add the number of the stations, at least the ones you used for Figure 9.
   - We added the station numbers in the google map too. The requested figure would be similar to figure 6(a) with fewer point in each plot and won't add much information. The correlation and RMSE of it is reported in figure 6(b). So, we decided not to include it.
2. Regarding the calibration, while I agree that the large number of stations reduce the bias, I am a bit concerned that this method can "mask" unreliable phase measurements. Take as an example the interferogram presented in Figure 5 (c), where most of the stations have very low coherence. If you do an averaging around the low coherence stations (considering homogeneity and a sufficient large number of pixels), phase is noisy and the expected value should be zero, which translates to 0 DeltaSWE. If now you add the mean DeltaSWE from the stations, you'll probably get something closer to the in situ value for that station, although the phase had no usable information. I'm not asking to change the methodology, just comment on this. Additionally, adding error bars with standard deviation in Figure 9 would be useful to assess the effect.
   - We have changed the calibration method so it would be easier for the readers. The point that you brought up is valid however, using the average of all stations remove that kind of errors as we have stations with different sort of temporal

coherence and SWE change. Note that we don't include stations with low temporal coherence for that average calculation. Having said that, we now are using 2 stations with very good correlation and temperature for the entire time series and it also should not have the problem you mentioned.
- Regarding the error bars to figure 9, I am not sure what you mean. In this figure we are just adding the delta SWE to get total swe at each observation. What std are you referring to?

3. See comments regarding Section 6.
- Done!
4. Perhaps it is a bit picky but be more consistent with coherence and correlation along the manuscript (e.g., lines 196, 205, F5 caption, 213… these could be coherence or temporal coherence). I am also missing some explanations why you neglect other sources of decorrelation and assume that the product is just the temporal component. Also, when discussing temperatures, I'd use 0°C over zero.
- We thought we captured all coherences! Thanks for pointing that out. We went through the paper again. Hopefully, now it is consistent.
- We changed all zero to 0°C.
- We added this explanation for figure 4: "Bottom row of figure 4 shows the coherence of the images in top row of figure 4. Interferometric decorrelation has different sources, such as temporal decorrelation, volume decorrelation, signal to noise ratio decorrelation, geometric decorrelation, .... The volume decorrelation is small due to relatively small Sentinel-1 perpendicular baseline. Temporal decorrelation is the dominant source of decorrelation. For the rest of this study, we assume the observed interferometric decorrelation is approximately the temporal coherence."

Specific Comments:

*Section 1:*

Good introduction. I am missing perhaps some mentions to co-polar phase difference for snow height estimation.
- Added: "The co-polar phase difference (CPD) between VV and HH polarization of X-band SAR acquisitions is used for estimating the depth of *fresh* snow (Leinss et al., 2014)."

Line 15: wouldn't mass be more precise than cover here?
- Changed it to "snow mass and cover". We need cover to know where it is and yes how much it is.

Line 50-56: perhaps add a few details. Phase center at the snow surface or the volume. I got a bit lost in this paragraph, when you say this method do you refer to the method for dry or wet snow? A sentence clarifying this could be useful.
- We shift the sentences to make it clearer: "The phase change of specularly reflected signals in Signals of Opportunity (SoOp) is shown to be strongly dependent on SWE changes for dry snow (Yueh et al., 2017, 2021; Shah et al., 2017). The theory behind using SoOp for SWE retrieval is similar to repeat pass interferometry that is explained in section 2. The advantage of this method is that the stratigraphy of the snow has little

impact on the SWE retrieval (Leinss et al., 2015; Yueh et al., 2017) similar to SWE retrieval explained in section 2. Using the long wavelength signal at P-band in SoOp is very helpful for addressing the loss of temporal coherence and phase unwrapping challenges of this method. However, the phase sensitivity to SWE changes decreases at lower frequencies. There has been very limited data showing the success of this method at P-band. Achieving high resolution is another challenge of this method. The phase change of specularly reflected signals in SoOp is dependent on snow depth change for wet snow (Yueh et al., 2017, 2021; Shah et al., 2017)."

Line 55: remove second "this method".
- Changed the sentence now, Done!

Line 57: remove small.
- Yes, done!

**Section 2:**

In Section 2.1 you could add that snow is a thermal insulator and it reduces temporal changes in the underlaying layers. Is this relevant for mountainous areas?
- We add this sentence: "On the other hand snow cover has a thermal insulation effect on the ground and underlaying layers (Gu et al., 2019). The insulation increases with the snow depth. Therefore, during the snow season we assume the ground remains frozen even when snow becomes wet. Hence, temporal decoherence from the ground is negligible."

Line 68: I think the idea is clear without "very" and "small"
- Fair enough, done!

Line 72: higher than what? or relatively high frequencies.
- Changed it to "relatively high frequencies such as C-band".

Line 75: Extend the equation to include . I guess makes some sense since you are focusing on SWE in the manuscript. Also details that snow permittivity depends on its density and can be approximated from the density and that the complex part is negligible for dry snow.
- I think the sentence is incomplete here. To include what? But your argument is right. We are focusing on dry snow that imaginary part of e is small and permittivity depends on mainly density.

Line 93 to 96: is this the correct place for explaining what you have done, or should it go in the introduction?
- We explained briefly in introduction section what our method is. However, we needed a section to explain what had been done and then explain what we did different or compliment to what have been done. I include this sentence to introduction section to highlight the difference over there too: "We show for the first time that SWE estimation using repeat pass interferometry performs well by using a *long* time series of Sentinel-1 interferometric data in winter 2021."

Line 101: two different times, for generality.
- Done!

Line 102 to 103: maybe make this idea about decorrelation more general, not just for vegetation.
- We changed it to: "However, the movements of the scatterers such as leaves and branches or sea ice particles decrease the temporal coherence."

Line 105: Consider changing faster sampling for shorter temporal baseline, sounds more connected to repeat pass interferometry.

- We changed it to: "Methods such as using two frequencies or shorter revisit time are used to overcome these problems."

*Section 3:*

Would be nice to explain that at the time of the study, Sentinel-1 operated a constellation of two satellites. You can mention that each of these satellites had a repeat pass of 12 days, but due to the orbit offset between them, the effective temporal baseline is 6 days.

- We add this sentence to the paper: "Sentinel-1 constellation includes Sentinel-1A and Sentinel-1B. These two satellites are in the same orbit with a $180^0$ orbital phasing difference. The revisit time for each of the satellites is 12 days. However, revisit time can get to 6 days if both satellites make observations."

I think Figure 2 deserved better visualization. The caption should mention that the green frame is the S1 image (as you did in Line 160), and what are red and green diamonds.

- We changed the caption to: "© Google Earth View (a) Google Earth View of Sentinel-1 path:71, frame:444 in Idaho.  (b) zoomed to the Sentinel-1 path:71, frame:444, shown by big green rectangle. Red boxes show the location of LIDAR data acquisition. The green diamonds show SNOTEL stations with ΔSWE error less than 2cm in the entire time series. The red diamonds show SNOTEL stations with ΔSWE more than 2cm in at least one observation in the time series. Yellow squares are SNOTEL stations 1 and 11 used for reference point."

Line 143: … at a central frequency of 5.405 GHz

- Done!

Line 145: comment that you can get S1 data at the Copernicus Open Access Hub :)

- Added: "The data are free and available through Alaska SAR Facility (ASF) or The Copernicus Data Hub distribution service."

Line 167: … for each of the SNOTEL stations.

- Done!

Line 209: change squares to diamonds?

- Good catch! Done!

Figure 3, (a): Change x label to "Day from 12/01".

- Done!

*Section 4:*

Line 185: are these sources of error general for all applications or just particular to snow?

- These are main sources of error for any application using Sentinel-1 data. We modified it to: "The main sources of error in the science and applications using Sentinel-1 repeat-pass interferometry are"

Line 195: be more specific specifying that it is data from SNOTEL stations and that this data is snow measurements.

- We changed it so it better captures your concern: "The temperature is also an important factor. Equation 1 is valid for dry snow ((Leinss et al., 2015), and we use near surface air

temperature above $0^0$ C as a metric that indicates wet snow in snow season. Any SNOTEL data with in situ near surface air temperature more than $0^0$ C is unreliable in our study."

Line 213: this idea about temporal coherence being a limitation of InSAR is repeated a couple of times in the paper. Consider removing this one.

- Agree, done!

Line 214-215: I don't understand the point of this sentence. Isn't it as short as possible? Or would you discuss limitations related to satellite capabilities?

- We removed that sentence as it is confusing. You are right. Our point is to use different data set to find the needed revisit time for future snow missions. We cannot afford as short as possible (every minute?) for a mission ☺

Line 221: water vapor content between passes.

- Done!

*Section 5:*

Line 239: is results the correct word here? Or LIDAR data/scan?

- Changed to LIDAR data.

Somewhere in this section you should state what is the window (number of pixels) used to extract the values of coherence and phase for the stations.

- We add this to this section: "As mentioned in section 3.1, the resolution of the Sentinel-1 InSAR data from Hyp3 is 80mx80m. We used a 10x10 multi-looks window of retrieved SWE and temporal coherence around the SNOTEL locations to reduce the speckle noise. Therefore, we compared the SNOTEL SWE with the 800mx800m retrieved SWE around the SNOTEL site. The heterogeneity of the environment such as vegetation cover, vegetation fraction, land type, and SWE distribution in the 800mx800m around the SNOTEL station affects our accuracy. We will analyze the effect of the heterogeneity of the environment on the SWE retrieval for SNOTEL stations in the future work of this study."

Line 255: change discovered to observed?

- Good suggestion, done!

Section 5.1.1: I would like to read some discussion about the points with error=0

- Some of the points were out of Sentinel-1 frame but in the square. Therefore, there was not data over there. We removed those stations since they were not actually in the frame. So, instead of 43 stations we have 31 stations in the new figures.
- There is no more zeros in figure 6(b) except for station 4 that we added this explanation: " Note that station 4 has just one observation with temporal coherence more than 0.35. That observation is the first observation with zero SWE change. Therefore, there is not enough points to calculate $\rho_i$. Hence, RMSE and correlation are zero."

Figure 7,c): are the dots placed on the reference acquisition?

- We add this sentence to clarify: "Note that the labels on x-axis show the first date of each interferometric observation."

One thing that comes to mind is if you can say something before the 12/25/20 acquisition (as it is the common one between the two interferograms you show) that could explain the noisy phase. E.g. wet snow, a storm…

- Thanks for the suggestion! We added this argument to the paper: "We observe that 4 out of 6 days between 12/19/20 and 12/25/20 (observation 4) are relatively warm including day 12/19/20. All 31 stations have temperature between $-7^0$C to $6^0$C at 6 am in those four days. The warm days cause a lot of melting and refreezing in those 4 days. Hence, we expect to have small temporal coherence and consequently noisier fringes. On the other hand, the temperature is relatively warm only on 12/26/20. The rest of the 5 days between 12/25/20 and 12/31/20 (observation 5) are mostly colder than $-7^0$C for all 31 stations. Therefore, higher temporal coherence and less noisier fringes.

Line 261: Do you mean observations 5 and 4? Otherwise, I don't think we know where stations 5 or 4 are…

- Yes, we meant observation not station, thanks for catching it, it is fixed now.

*Section 5.1.2:*

Line 269: As mentioned in Section 5.1.1? or in Section 4? But check this sentence if I understood correctly you use DeltaSWE values when temporal coherence is below 0.35 and temperature higher than 0°. So, does this mean all values?

- It was explained better in section 4 but also in section 5.1.1. On the other hand, there was a typo here. We only used data with high temporal coherence and temperature lower than $0^0$C. Here is the new sentence: "However, as mentioned in section 4 and 5.1.1, we only used $\Delta$SWE values with temporal coherence more than 0.35 and temperature less than 0∘C." We discarded high temperature or low coherence data.

Please indicate that you set SWE_t1 as 0. Some stations have snow already for the first acquisition. I think you need a comment on this.

- We add this sentence to clarify: "Note that the SWE($t_i$+1) is measured compared to SWE($t_1$). For simplicity, we assume the SWE at time $t_1$ is equal to zero."

Figure 9: Do the timeseries expand the same time span. I think they do but the x axis is different for the three subfigures… Also, drop some of the decimal precision? See comment about std for each acquisition.

- We changed the time to observation number with viable retrieval (coherence >0.35 and T<) to avoid confusion. We explained it in the text: "We had 18 observations for the entire time series. Discarding some observation due to low temporal coherence or high temperature, changes the time series length. As seen in figure 8, we keep all 18 observations for station 20 but only 15 observations for station 12."

Line 295: could you be a bit more precise about how a station is labeled as red or green? What is the threshold for it? I see this is explained in the caption of Figure 10, but shouldn't it be in the text also?

- We add this threshold to the text and corrected in the caption: "The SNOTEL sites are shown by small diamonds in figure 2(b). The green small diamonds have a total SWE error less than 2 cm in the entire time series, similar to stations 12 and 30. The red diamonds have a total SWE error more than 2cm, similar to station 20. However, the retrieved SWE has a similar pattern as in situ SWE."

*Section 5.2:*

I think the analysis here could be extended. Is there anything that is increasing the error like steep slopes or vegetation (although from figure 2 seems there is not much vegetation)? Could you add some comments about coherence for those images? Was it generally conserved? Another point just out of curiosity, are the results similar if you compare the LIDAR map to S1 SWE map from 03/13/21? Looking to Figure 3.b) does not seem there were much SWE accumulated between the pass dates.

- Thanks for the suggestions! The section 5.2 is expanded now. We are working on a second paper to better discuss the effect of different parameters such as vegetation, slope, temperature, … . It is beyond the scope of this work. However, we added two figures to show the mean of temporal coherence for these two sites and how it affects the correlation.
- We also added a figure showing the correlation between LIDAR snow depth and retrieved SWE for different observation dates.

Line 303: Add the complete date of the LIDAR scan.

- We add the flight date: "The terrain DEM is measured by LIDAR sensor during September 2021. The DEM is used to measure the snow depth using the LIDAR data collected on 3/15/2021."

Line 305: isn't the closest date the 03/13/21?

- Yes, but we wanted to cover the entire time series. We can go with 3/13 or 3/19. The results doesn't change significantly.

Line 311: total SWE accumulated rather than change?

- Corrected!

Perhaps indicate in the text that in Figures 11 and 12, (c) what does color represent?

- We added: "The 2D-histograms of these two images are shown in figure 10(c) and 11(c) where x- and y-axis show the LIDAR snow depth and Sentinel-1 retrieved SWE, respectively. The colors in part (c) shows the $10^{number\ of\ cells}$ with LIDAR snow depth x in part(a) and InSAR SWE y in part (b).

**Section 6:**

I think this section is a bit short and does not discuss important aspects of the work. You could emphasize that the total SWE vs LIDAR was done accumulating the contribution from many interferograms, and the great spatial match. This to me is a truly impressive result. On the other hand, you should also need to discuss the calibration strategy, in my opinion this plays a crucial role in your results you cannot overlook commenting on it.

- We expand this section more to cover referencing problem and LIDAR results better: "We showed that retrieved ΔSWE using Sentinel-1 is highly correlated (0.8) with in situ values, with an RMSE of 0.93cm. For reference point of interferometric phase, we used two in situ stations with temporal coherence more than 0.35 and temperature less than 0∘C for the entire time series. We subtracted the difference between the average of in situ and retrieved ΔSWE of these two stations from retrieved values to calibrate the retrieved ΔSWE. The ΔSWE RMSE error is less than 2 cm for all stations and less than 1 cm for most stations. The correlation between retrieved and in situ ΔSWE is more than 0.6 for most stations. The ΔSWE retrieval performance degrades for days with small ΔSWE. We demonstrated that low temporal coherence not only degrades the SWE retrieval performance, but also the unwrapping algorithm performance. We showed that big

melting events between two Sentinel-1 acquisitions make the interferometric fringes noisy and unwrapping algorithm challenging.

The retrieved total SWE has less than 2cm RMSE compared with in situ values for 9 stations and more than 2 cm for 14 stations.

The highlight of the results of this study is the similarity between two independent measurements, retrieved SWE using Sentinel-1 data and LIDAR snow depth data. We used Sentinel-1 data between 12/01/20 to 03/19/21 to retrieve ΔSWE time series. By adding the entire time series of ΔSWE, we calculated the total SWE on 03/19/21. Total retrieved SWE using Sentinel-1 interferometric data and LIDAR snow depth images over two regions in Idaho show similar patterns and are correlated by more than 0.47. We showed that the correlation is higher for regions with higher temporal coherence in Banner Summit. … It is also shown in this study that melting due to warm temperature reduces the temporal coherence and the performance of unwrapping algorithm."

Line 324: Not completely sure about this. Definitely short temporal baselines will help with coherence but if between passes there is a snowstorm most likely coherence will drop significantly.

- Analyzing the temporal coherence in this study shows that temporal coherence decay exponentially with swe change. So, you are right that big snow storm decrease the temporal coherence but with smaller temporal baseline the swe change also decreases. As time is correlated to swe change in the snow season.

Technical Comments:

Line 191: Section 5.12, I think

- We don't have section 5.12. I am not sure what you mean here. We used all the data in section 5.2 even the ones with low coherence and high temperature.

Line 196: Section 5.1.2, I think

- I am not sure. We meant in section 5.2. Note we removed section 5.1.2 in the new version of the paper. "We used all the data in section 5.2, including the data with low coherence and high temperature."

Line 214: InSAR (capitalization)

- We removed this sentence from the new version of the paper.

Figure 7. (a): Path:71. Why 2021? Should not be from 2020 to 2021?

- Corrected: "Retrieved ΔSWE using Sentinel-1 interferometric phase versus in situ ΔSWE for all the stations with temporal coherence more than 0.35 for the entire Sentinel-1 time series from December 2020 to March 2021."

Line 256: InSAR(capitalization)

- Corrected!

Line 333: NISAR acronym already introduced.

- Removed, thanks!

---

## Referee Report (RR1)

After the revision my concerns have been cleared. I think the new content added is valuable and has improved the article. Congratulations once again to the authors. I have some minor comments to add:

Regarding my request about Figure 9. What I was asking for was adding some information about the spatial variability in the window you used for comparing the retrieval with the SNOTEL stations. I think that error bars containing the std within the window could add some useful information. After the revision it is clear how you addressed the calculation of the values of DeltaSWE, so I see no need of including this.

Regarding my comment about Line 75, it seems the equation I wrote in the comments disappeared as I copied into the portal… I was asking for an explicit expansion of DeltaDepth to DeltaDepth = DeltaSWE*rho_water/rho_snow, but I see it has been addressed in the revision.

Figure 2. I think figure has improved quite a bit! Another idea that may help improve visualization is assigning different colours to the boxes of Banner Summit and More Creek. Now is not easily identifiable. I see this is explained in line 364 but perhaps it could be clearer.

Figure 6.c: in the text you refer to observations but in this figure, you have dates. Is a bit confusing.

Line 304: I don't completely agree with this claim. An accumulation of 0.5 cm of SWE should induce approximately pi/3 radians of phase. I think this is a sensible amount of accumulation at C-band. Can you specify if the value of the average dSWE is in each interferogram or along all the interferograms?

Figures 10, 11 and 12: Consider adding a point with the location of the SNOTEL stations. Feel free to ignore this idea!

Figure 13 (b): The correlation for observation 1 is quite high although it's at the beginning of the winter. Can you provide an explanation of why this is observed?

Line 420: I don't fully understand this sentence. Do you mean that the retrieval degrades for interferograms with small dSWE?

Section 6: I think you could discuss how the calibration strategy may have help overcome some limitations regarding phase ambiguity. In case you consider this is of relevance.

Technical Comments:

Figure 2. Caption: "The red diamonds show SNOTEL stations with ΔSW E more  than 2cm in at least one observation in the time series."

Line 241: comma? (, using the average…)

Line 251: geometric

Line 252: is negligible.

Line 355: isn't it clearer "two of the stations"?

Line 394: add °C.

Line 399: LIDAR

Line 404: missing blank space?

Line 439: double dataset in the sentence(?)

---

## Author Response (AR2)

We would like to thank the reviewers for the detailed and very useful review. We appreciate the reviewer's time and effort. We believe the manuscript is now in a much better shape. We hope this manuscript help the community in future approaches for SWE retrieval missions. The changes are relatively minor compared to last revision.

Please find our responses below.

**RC1, Silvan Leinss**

Review, Revision 1 of TC-2023-95, Snow Water Equivalent Retrieval Over Ideaho, Part A: Using Sentinel-1 Repeat-Pass Interferometry

general comment:
The authors improved the manuscript significantly and I am satisfied with almost all comments.
- Thanks for your useful review. It improves the paper a lot.

Below I have a couple of minor issues that need to be fixed before publication. Specifically, I do not agree that the authors claim to provide a "more general" method than what has been suggested in Leinss(2015). Below (comment for line 140) I compare the validity range of the two approximations and show that the suggested equations (4) is valid for a wider incidence angle range compared to Eq. (17) in Leinss (2015). However, the suggested equation (4) is limited to realistic densities of seasonal while the equation (17) in Leinss (2015) is valid for all densities up to solid ice. The suggested equation is therefore slightly better suited for the analysis of S1 data, but I would not call it "more general".
- We changed it as mentioned below.

I also like to point out that the Equation from Gunneriussen, that relates snow depth, permittivity to phase, and that has been approximated to derrive SWE, is only valid when SWE is considered perpendicular to the local slope, not as a vertical water column as commonly used. I guess, the authors approximate the vertical water column with their approximation and neglect the conversion of SWE perpendicular to the local slope to the vertical water column. If this is the case, it needs to be clearly stated.
- We are also estimating the SWE perpendicular to the surface. As mentioned below we clarify it more in the paper now.

specific comments:

Abstract:
l.9: It would be helpful to provide an approximate value of the total SWE (or some order of magnitude) to have a comparison to the given RMSE and max. Error.
- We add this sentence: "The SWE change measurements vary between -5.3cm to 9.4cm over the entire time series and all the in situ stations."

Could you provide a sentence of conclusion or recommendation from your study in the abstract? What would improve the performance? Lower frequency? Higher repeat-pass intervals? something else?
- We added this sentence: "Higher frequency such as L-band improves the temporal coherence and SWE ambiguity. SWE retrieval using C-band Sentinel-1 data is shown to

be successful, but faster revisit is required to avoid low temporal coherence. Global SWE retrieval using radar interferometry will have a great opportunity with the upcoming L-band 12-day repeat pass NISAR data and the future 6-day repeat pass ROSE-L data."

Introduction:
l. 51: Could you add the imaging method used to map snow pack properties? Was a nadir-looking altimeter used? Or side-looking real aperture radar or a SAR system?
- We add this sentence to the manuscript to make it clear: "The system was normally deployed nadir looking and was a real aperture radar system."

l. 50-54: This paragraph needs to be improved. According to your response letter, the reason that "FMCW systems requires large bandwidths is to avoid ambiguities". "Something that we do with prf or pulsed signal". In the manuscript, line 52, you write "These [FMCW] sensors need a wide bandwidth for large distance of a spaceborne mission with high resolution". This sentence is incomplete. I cannot follow your argumentation here, neither in the response letter, nor in the manuscript. The cited paper (Marshall and Koh, 2008) discuss FMCW radars in the context of use the term "large distance" in the context of the length of snow transects of a few hundreds of meters. They use the term "high resolution" in the context of vertical slant range resolution, i.e. they use FMCW radar as radar sounders with maximum slant-range distances of a few tens of meters.
FMCW systems estimate distances from the frequency difference $f\_b$ ("beat frequency") between the echo and the local instantaneous chirp frequency. The echo returns after a time delay $\tau = 2R/c$ (R: slant range, c: speed of light). With that, the frequency difference is $f\_b = \beta*\tau$ with the frequency change rate within the chirp of $beta = B*PRF$ (B: bandwidth, PRF: pulse rep. freq.). By decreasing bandwidth B or PRF the frequency differences $f\_b$ decreases. The minimum and maximum distance within the swath determines the bandwidth of $f\_b$ that needs to be sampled by the ADC. This bandwidth, however, can be significantly smaller than the bandwidth B of the transmitted chirp that determines the slant range resolution. So, frequency allocation in space limits the possible slant-range resolution (for all radar systems, not only FMCW).
- The FMCW systems have high range resolution at least for SWE on the order of few cm to detect different layers in the snow. The required bandwidth for that kind of resolution is on the order of GHz. The spaceborne allocation for NISAR is 80MHz and not more than that. Therefore, it is not feasible to have spaceborne FMCW for snow. We tried to explain this better now in the paper: "The resolution of FMCW system for SWE application is in cm scale. In order to achieve such high resolution, the bandwidth should be in GHz scale. Due to limitation on frequency bandwidth allocation of a spaceborne active sensor (Tao et al., 2019), FMCW systems cannot be used in spaceborne missions for global coverage due to their wide bandwidth."

l.62: "achieving high resolution is another challenge of this method": do you mean the SWE retrieval method or the data from SoOp? You have not explained what kind of imaging method SoOp is, but I guess, that the limited resolution of this method is the local data (like GPS) or not focused SAR. - but not the SWE retrieval method.
- Yes, the challenge is for SoOp data that is not focused. However, the data will be used to estimate SWE. So, the SWE will have low resolution too. We changed the sentence to this:" Achieving high resolution for SoOp data is another challenge."

l. 68, 71, 106, 428: "for the first time": For sure, you did that for the first time, that's the reason why its worth to publish your results in a scientific journal for "original research". IMHO, you can write "for the first time" in a newspaper, not scientific journal. Usually, when I read such phrases it does not speak for the quality of the paper. The results itself should highlight the quality and impact, not the emphasis that this is the very very first time.

- Thanks for mentioning that. I didn't realize I have used it 3 times in the paper. I guess I was excited about the method working. However, the reason we are saying this is to make it easier for the researchers to follow this kind of work in future. I think mentioning that makes it easier for the scientists to know what in this work is different from other works. To me Leinss et al. 2015 was the beginning of this method in small region and it was easier to follow what has been done afterward. I was lucky to be introduced to this work soon enough and I didn't need to do lots of literature search to realize that. Beside I have seen using this phrase in other people's work. Having said that I removed three instances and just kept this phrase for the conclusion section.

Section 2:

You use a very large range of incidence angles to estimate SWE. Do you estimate the SWE perpendicular to the local slope or the SWE as a vertical column even in terrain with a local slope? The equation of Gunneriussen, and therefore all derrived approximations, are made for horizontal terrain or, equally, the SWE perpendicular to the local slope - which is different from the vertical water column of a snow pack. I assume, you make the assumption that the terrain is not very steep so that both are approximately the same. Please specify.

- The terrain is assumed to be parallel to snow layer. That is the assumption we made which I think is a reasonable assumption. The SWE change that we measure is also perpendicular to local terrain as you mentioned similar to all the other approximations. It is not vertical water column of snowpack. In that sense we need to find the incidence angle to the local terrain with any sort of slope. That is why we looked at the local incidence angle (with large range depending on the local slop) not the general incidence angle for a flat train which changes from 20 to 50 or so.

140: "we try to make a more generalized approximation": Could you specify, here, what is more generalized? Later, you wrote that you have a single equation (4), that depends on \theta, for the assumption of snow density between 0.15 and 0.45, and an incidence angle between 0 and 80° (note: line 158, you specify the error only up to 70°). According to Fig. 1c, the maximum errors relative to, i guess, the equation from Gunneriussen are below 15% for densities between 0.15 and 0.45 and incidence angles between 0 and 70°. Limiting yourself to 10..60° reduces the maximum error to 5% (you won't analyze data at 0° incidence angle due to layover). Comparing that to Equation (17) in Leinss(2015): With the simplification of alpha = 1 the maximum error for the same density and incidence angle range is 12.5% (at incidence angles of 60°). The error increases for larger incidence angles (gray contours in Fig. 9, right). However, equation (17) in Leinss (2015) holds for all densities up to solid ice, but is limited to incidence angles of below, let's say, 60°. Your equation (4) is limited to common snow densities between 0.15 and 0.45 g/cm^3 but provides a lower error at large incidence angles (let's say, up to 70 or 80°). So both cover the same two dimensions (density, incidence angle) to slightly different extent. Both equations depend on incidence angle (which is usually known) and both do not depend on density (which makes both equations very valuable for SWE retrival). The "Leinss-Equation" is more accurate (I would not say more general) for high densies of snow up to solid ice, while the

"Oveisgharan-Equation" is more accurate (again, I would not say more general) for incidence angles above 50 or 60 degree and therefore probably better suited for an analysis of Sentinel-1 where such large incidence angles are likely to occur on slopes facing off the radar. So please clearly describe the differences in limitation for density and incidence angles as detailed above. I do not agree with the claim your equation is more general. I would agree that it is more general, if the equation would covers another variable, e.g. if it would be independent of the liquid water content, or if it would be at least valid for the same density range but valid for an significantly increase incidence angle range.

- I only want to mention the reason we didn't use the snow densities more than 0.5 is that the terrestrial snow density normally ranges between these values. As you can see in figure 1.c the error gets large for small density. However, I agree that Leinss's approximation is valid for wider range of snow density. We changed the text in paper and we hope this captures your concern: "The approximation is limited to a smaller range of incidence angle than Sentinel-1 incidence angle. However, Leinss et al. approximation applies to a wide range of snow density up to solid ice density. Due to the wide range of Sentinel-1 incidence angle in a frame, we tried to make a more accurate approximation for a wider range of incidence angles and snow densities limited to terrestrial snow.

In you response letter, for line 139, you write "I think our equation is applicable to a wide range of snow densities". That is not true, see above. It is only applicable to a wide range of incidence angles which makes it convenient for S1.

- We change that as mentioned above

l.145 vs. l.149: C(theta, rho) is specified with two different equations. Use a different symbol, e.g. C', for the approximation in line 149 and subsequent references to it.

- We changed it to C^ for L149.

149: "as seen in fig. 1(a), the line intercept is very close to zero": It is not clear what Fig. 1a shows (see comment above), but as the shown lines have a small curvature, these are likely C(theta, rho) from line 145. As this is not a linear function, you can't talk about line intercepts. Why not adding two dashed lines, defined by the linear function C(theta, rho) from line 149, and concluding from that that C(theta, rho) from 145 is very close to the linear function (C in l.149) without y-axis intecept.

- We added the fitted line to figure 1.a. eventhough the error of this fitting is also shown in figure 1.c. We also added to text:" The blue and red dashed lines show $\hat{C}(\theta,\rho)$ at incidence angles equal to zero and 70, respectively. As seen in figure 1(a), the fitted line with zero intercept is a good approximation. The Zero intercept approximation is essential to retrieve ΔSW E independent of snow density."

Fig. 1b: Please show the fitted values of the line slope A(theta), together with the polynominal approximation A from eq. (3).

- Done!

Fig. 1c: Please specify the caption how the error was calculated. Line 157 provides some hint.

- We add the formula to the caption too.

l.198: You specify the height of your stations between 3200m and 9520m. Do you mean feet? Maybe, you should convert feet to meters?

- Thanks for catching that, we changed it to: "975m to 2902m."

l.231: "Any SNOWTEL data with ... is unreliable in our study": Do you mean "Any SWE retrived from S1 for locations where SNOWTEL data indicates positive in-situ data is considered unreliable"?

- Correct! We changed it to: "Any retrieved SWE with SNOTEL near surface air temperature more than 0 is unreliable in our study."

Conclusion:

l. 411: "We showed that retrieved Delta SWE..." Could you specifiy if you mean the SWE between two acquisitions or the total DeltaSWE (Eq. 5) during the dry snow period?
- We mean delta SWE between two acquisitions. Equation 5 is called total SWE in this study. We make it clearer in the text too: "We showed that retrieved Delta SWE between two consecutive Sentinel-1 observations"

technical corrections:

l. 12: I do not think that the phase-unwrapping algorithms themselves degrade, rather their performance or results. I would end the previous sentence with something like "... because of unsuccessful phase-unwrapping."
- We changed it to:" We also show that the performance of the phase unwrapping algorithm degrades in regions with low temporal coherence."

43: "restricted to passive retrival errors" - do you mean "restricted/limited by passive retrival errors?"
- Changed the "restricted to" to "limited by".

62: check grammar of this sentence. I don't get the meaning.
- We removed this sentence. The relationship between phase and snow depth in wet snow is not relevant to this work. As you mentioned it needs more explanation.

90: to the ground and the scattering.. -> so that the scattering
- Done!

**RC2, Jorge Jorge Ruiz**

After the revision my concerns have been cleared. I think the new content added is valuable and has improved the article. Congratulations once again to the authors.
- Thanks so much for reviewing our work. Your feedbacks really improved the work and paper.

I have some minor comments to add:
Regarding my request about Figure 9. What I was asking for was adding some information about the spatial variability in the window you used for comparing the retrieval with the SNOTEL stations. I think that error bars containing the std within the window could add some useful information. After the revision it is clear how you addressed the calculation of the values of DeltaSWE, so I see no need of including this.
- I see, I am glad the revision makes it clear.

Regarding my comment about Line 75, it seems the equation I wrote in the comments disappeared as I copied into the portal… I was asking for an explicit expansion of DeltaDepth to DeltaDepth = DeltaSWE*rho_water/rho_snow, but I see it has been addressed in the revision.
- Yes, it is now added correctly in the paper, thanks for clarifying.

Figure 2. I think figure has improved quite a bit! Another idea that may help improve visualization is assigning different colours to the boxes of Banner Summit and More Creek. Now is not easily identifiable. I see this is explained in line 364 but perhaps it could be clearer.

- I changed the color of Banner summit and explained it in the text.

Figure 6.c: in the text you refer to observations but in this figure, you have dates. Is a bit confusing.

- It was difficult to mention the date each time. However, we added the date of observations to the text. For instance, "observations 1, 2, 7, 9, 15, and 16 (first date of 12/01, 12/07, 01/06, 01/18, 03/07, and 03/13)".

Line 304: I don't completely agree with this claim. An accumulation of 0.5 cm of SWE should induce approximately pi/3 radians of phase. I think this is a sensible amount of accumulation at C-band. Can you specify if the value of the average dSWE is in each interferogram or along all the interferograms?

- This is correct. However what we mean is that for instance the average of swe change on day 2 is less than 0.5cm among all 31 stations (the average of swe change between 31 stations on day 1, 2, 7, 9, 15, and 16 is 0, 0, 0.4, 0.3, 0.2, 0.4cm, respectively) . So, even though the mean is less than 0.5cm, most stations have swe change of close to zero. And the correlation between retrieved and in situ is not meaningful. We added this sentence to make it more clear in the text. "Among the observation dates with correlation less than 0.35 (observation 1, 2, 4, 7, 9, 15, 16, and 17), observations 1, 2, 7, 9, 15, and 16 (first date of 12/01, 12/07, 01/06, 01/18, 03/07, and 03/13) have very small snow accumulation (the average $\Delta$SW E is less than 0.5cm with $\Delta$SWE close to zero for most stations)." There are couple of stations with relatively higher swe change but with most stations close to zero swe change, we won't expect a good correlation. Average swe change less than 0.5cm is sort of an indication that there hasn't been snow storm in many stations. We sort of take that as an indication of where/when we expect to a good measurement. However, for stations with large swe change for those dates and even 0.5cm we should be able to see phase change. What we mean here is that the correlation for all stations won't be meaningful. For your reference we include the figure showing the SWE change for these dates below

[Figure]

Figures 10, 11 and 12: Consider adding a point with the location of the SNOTEL stations. Feel free to ignore this idea!

- We added the SNOTEL station to figures 10(a) and 11(a) and explained in the text: "As shown in figure 2(b), Banner Summit covers SNOTEL 2 and Mores Creek covers

SNOTEL 21. These two SNOTEL stations are shown by diamonds in figures 10(a) and 11(a)."

Figure 13 (b): The correlation for observation 1 is quite high although it's at the beginning of the winter. Can you provide an explanation of why this is observed?

- That is a very good observation. Note station 21 is relatively in the low LIDAR snow depth region. Looking at the retrieved SWE for the first date we see snowstorm in the high topography region. We believe, there has been snow in the first day in high elevation regions and that is what we see in this image. We added this sentence: "Although ΔSWE for station 21 is zero for the first observation, the correlation between LIDAR snow depth on 03/19/21 and total SWE on the first observation is relatively high, as seen in figure 13(b). As shown in figure 11(a), station 2 is in the relatively low snow depth region. We believe there has been a snowstorm in the high-altitude region of Mores Creek. The high correlation is simply the correlation between LIDAR data on 03/19/21 and the first snowstorm in Mores Creek."

Line 420: I don't fully understand this sentence. Do you mean that the retrieval degrades for interferograms with small dSWE?

- What we meant is that the correlation between retrieved dswe and in situ dswe is small for days with small dswe. We clarified it more in the text:" Interferograms with small average of in situ ΔSWE show low correlation between retrieved and in situ ΔSWE."

Section 6: I think you could discuss how the calibration strategy may have help overcome some limitations regarding phase ambiguity. In case you consider this is of relevance.

- You are right! However, we didn't show that in the paper and it is one of our future work to compare different reference point methods. We added this sentence: "We think using in situ station as the reference point helps reducing the phase ambiguity error, at least locally, compared to other methods for referencing the interferometric images. If the temporal coherence is large enough for the entire image to reduce the phase unwrapping error, using the in situ ΔSWE as the reference point reduces the phase ambiguity error in a larger region. Using a snow-free point or snow-free corner reflector as the reference point, cannot address the phase ambiguity in regions with deep snow."

Technical Comments:

Figure 2. Caption: "The red diamonds show SNOTEL stations with ΔSW E more less than 2cm in at least one observation in the time series."

- Corrected: "The red diamonds show SNOTEL stations with ΔSW E error more than 2cm in at least one observation in the time series."

Line 241: comma? (, using the average…)

- I checked online, it seems either or doesn't need comma!

Line 251: geometric

- Corrected, thanks!

Line 252: is negligible.

- Done!

Line 355: isn't it clearer "two of the stations"?

- I changed it to "two stations".

Line 394: add °C.

- Done!

Line 399: LIDAR

- Fixed!

Line 404: missing blank space?

- Done!

Line 439: double dataset in the sentence(?)

- Corrected: "With the L-band NISAR launch coming next winter, the new dataset would be a great opportunity for global SWE retrieval."